# RGI: ROBUST GAN-INVERSION FOR MASK-FREE IMAGE INPAINTING AND UNSUPERVISED PIXEL-WISE ANOMALY DETECTION

**Shancong Mou**[*], **Xiaoyi Gu**[*], **Meng Cao**[†], **Haoping Bai**[†], **Ping Huang**[†], **Jiulong Shan**[†], **Jianjun Shi**[*]

[*]Georgia Institute of Technology, [†]Apple

{shancong.mou,xiaoyigu}@gatech.edu, jianjun.shi@isye.gatech.edu
{mengcao, haoping_bai, huang_ping, jlshan}@apple.com

## ABSTRACT

Generative adversarial networks (GANs), trained on a large-scale image dataset, can be a good approximator of the natural image manifold. GAN-inversion, using a pre-trained generator as a deep generative prior, is a promising tool for image restoration under corruptions. However, the performance of GAN-inversion can be limited by a lack of robustness to unknown gross corruptions, i.e., the restored image might easily deviate from the ground truth. In this paper, we propose a Robust GAN-inversion (RGI) method with a provable robustness guarantee to achieve image restoration under unknown *gross* corruptions, where a small fraction of pixels are completely corrupted. Under mild assumptions, we show that the restored image and the identified corrupted region mask converge asymptotically to the ground truth. Moreover, we extend RGI to Relaxed-RGI (R-RGI) for generator fine-tuning to mitigate the gap between the GAN learned manifold and the true image manifold while avoiding trivial overfitting to the corrupted input image, which further improves the image restoration and corrupted region mask identification performance. The proposed RGI/R-RGI method unifies two important applications with state-of-the-art (SOTA) performance: (i) mask-free semantic inpainting, where the corruptions are unknown missing regions, the restored background can be used to restore the missing content. (ii) unsupervised pixel-wise anomaly detection, where the corruptions are unknown anomalous regions, the retrieved mask can be used as the anomalous region's segmentation mask.

## 1 INTRODUCTION

When trained on large-scale natural image datasets, GAN (Goodfellow et al., 2020) is a good approximator of the underlying true image manifold. It captures rich knowledge of natural images and can serve as an image prior. Recently, utilizing the learned prior through GANs shows impressive results in various tasks, including the image restoration (Yeh et al., 2017; Pan et al., 2021; Gu et al., 2020), unsupervised anomaly detection (Schlegl et al., 2017; Xia et al., 2022b) and so on. In those applications, GAN learns a deep generative image prior (DGP) to approximate the underlying true image manifold. Then, for any input image, GAN-inversion (Zhu et al., 2016) is used to search for the nearest image on the learned manifold, i.e., recover the $d$-dimensional latent vector $\hat{z}$ by

$$\hat{z} = \arg\min_{z \in \mathbb{R}^d} L_{rec}(x, G(z)), \tag{1}$$

where $G(\cdot)$ is the pre-trained generator, $x$ is the input image, and $L_{rec}(\cdot, \cdot)$ is the loss function measuring the distance between $x$ and the restored image $\hat{x} = G(\hat{z})$, such as $l_1, l_2$-norm distance and perceptual loss (Johnson et al., 2016), or combinations thereof. However, this approach may fail when $x$ is grossly corrupted by unknown corruptions, i.e., a small fraction of pixels are completely corrupted with unknown locations and magnitude. For example, in semantic image inpainting (Yeh et al., 2017), where the corruptions are unknown missing regions, a pre-configured missing regions' segmentation mask is needed to exclude the missing regions' influence on the optimization procedure. Otherwise, the restored image will easily deviate from the ground truth image (Figure 1).

For another example, in unsupervised anomaly detection (Schlegl et al., 2017), where the anomalies naturally occur as unknown gross corruptions and the residual between the input image and the restored image is adopted as the anomaly segmentation mask, i.e., $x - G(\hat{z})$, such a deviation will deteriorate the segmentation performance. However, the assumption of knowing a pre-configured corrupted region mask can be strong (for semantic inpainting) or even invalid (for unsupervised anomaly detection). Therefore, **improving the robustness** of GAN-inversion under unknown *gross* corruptions is important.

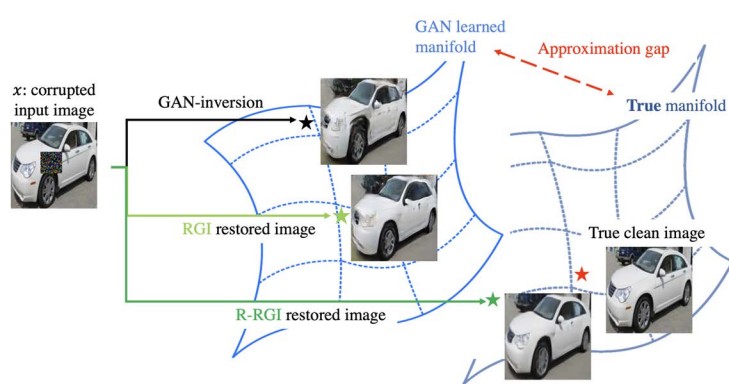

Figure 1: Inverting a corrupted test image in Stanford cars dataset (Krause et al., 2013) (i) the GAN-inversion restored background can significantly deviate from the ground truth; In contrast, the RGI method achieves a robust background restoration under unknown gross corruptions; (ii) due to the GAN approximation gap, the true clean image may not live on the GAN learned image manifold; R-RGI can further fine tune the learned manifold toward the true image manifold.

Another problem is the GAN approximation gap between the GAN learned image manifold and the true image manifold, i.e., even without corruptions, the restored image $\hat{x}$ from Equation 1 can contain significant mismatches to the input image $x$. This limits the performance of GAN-based methods for semantic inpainting and, especially for unsupervised anomaly detection since any mismatch between the restored image and the input image will be counted towards the anomaly score. When a segmentation mask of the corrupted region is known, such an approximation gap can be mitigated by fine-tuning the generator (Pan et al., 2021). However, adopting such a technique under unknown gross corruptions can trivially overfit the corrupted image and fail at restoration. Therefore, **mitigating GAN approximation gap** under unknown *gross* corruptions is important.

To address these issues, we propose an RGI method and further generalize it to R-RGI. For any corrupted input image, the proposed method can simultaneously restore the corresponding clean image and extract the corrupted region mask. The main contributions of the proposed method are:

**Methodologically**, RGI improves the robustness of GAN-inversion in the presence of unknown gross corruptions. We further prove that, under mild assumptions, (i) the RGI restored image (and identified mask) asymptotically converges to the true clean image (and the true binary mask of the corrupted region) (Theorems 1 and 2); (ii) in addition to asymptotic results, for a properly selected tuning parameter, the true mask of the corrupted region is given by simply thresholding the RGI identified mask (Theorem 2). (iii) Moreover, we generalize the RGI method to R-RGI for meaningful generator fine-tuning to mitigate the approximation gap under unknown gross corruptions.

**Practically** (i) for mask-free semantic inpainting, where the corruptions are unknown missing regions, the restored background can be used to restore the missing content; (ii) for unsupervised pixel-wise anomaly detection, where the corruptions are unknown anomalous regions, the retrieved mask can be used as the anomalous region's segmentation mask. The RGI/R-RGI method unifies these two important tasks and achieves SOTA performance in both tasks.

## 2 RELATED LITERATURE

**GAN-inversion** (Xia et al., 2022a) aims to project any given image to the latent space of a pretrained generator. The inverted latent code can be used for various downstream tasks, including GAN-based image editing (Wang et al., 2022a), restoration (Pan et al., 2021), and so on. GAN-inversion can be categorized into learning-based, optimization-based, and hybrid methods. The

objective of the learning-based inversion method is to train an encoder network to map an image into the latent space based on which the reconstructed image closely resembles the original. Despite its fast inversion speed, learning-based inversion usually leads to poor reconstruction quality (Zhu et al., 2020; Richardson et al., 2021; Creswell & Bharath, 2018). Optimization-based methods directly solve a latent code that minimizes the reconstruction loss in Equation 1 through backpropagation (which can be time-consuming), with superior image restoration quality. Hybrid methods balance the trade-off between the aforementioned two methods (Xia et al., 2022a). There are also different latent spaces to be projected on, such as the $\mathcal{Z}$ space applicable for inverting all GANs, $m\mathcal{Z}$ space (Gu et al., 2020), $\mathcal{W}$ and $\mathcal{W}^+$ spaces for StyleGANs (Karras et al., 2019; Abdal et al., 2019; 2020) and so on. All these methods do not have explicit robustness guarantees with respect to gross corruptions [1]. To improve the robustness of GAN-inversion, MimicGAN (Anirudh et al., 2018) uses a surrogate network to mimic the unknown gross corruptions at the test time. However, this method requires multiple test images with the same corruptions to learn a surrogate network. Here, we focus on developing a robust GAN-inversion for optimization-based methods, projecting onto the most commonly used $\mathcal{Z}$ space, with a provable robustness guarantee. The proposed method can be applied to a single image with unknown gross corruptions, and has the potential to be applied to learning-based as well as hybrid methods, even for different latent spaces, to increase their robustness.

As mentioned in the Introduction, DGP plays an important role in corrupted image restoration. GAN-inversion is an effective way of exploiting the DGP captured by a GAN. Therefore, GAN-inversion gains popularity in two important applications of corrupted image restoration: semantic image inpainting and unsupervised anomaly detection. (Comprehensive reviews on semantic image inpainting and unsupervised anomaly detection are provided in Appendix A.1 and A.2.)

**Mask-free Semantic inpainting** aims to restore the missing region of an input image with little or no information on the missing region in both the training and testing stages. GAN-inversion for semantic inpainting was first introduced by Yeh et al. (2017) and was further developed by Gu et al. (2020); Pan et al. (2021); Wang et al. (2022b); El Helou & Süsstrunk (2022) for improving inpainting quality. Current GAN-inversion based methods have the advantage of inpainting a single image with arbitrary missing regions, without any requirement for missing region mask information in the training stage. However, they do require a pre-configured missing region mask for reliable inpainting of a corrupted test image. Otherwise, the restored image can deviate from the true image. Moreover, the pre-configured corrupted region mask is also the key in mitigating the GAN approximation gap (Pan et al., 2021) through generator fine-tuning. Such a pre-configured corrupted region mask requirement hinders the application of GAN-inversion in mask-free semantic inpainting.

**Unsupervised pixel-wise anomaly detection** aims to extract a pixel-level segmentation mask for anomalous regions, which plays an important role in industrial cosmetic defect inspection and medical applications (Yan et al., 2017; Baur et al., 2021). Unsupervised pixel-wise anomaly detection extracts the anomalous region segmentation mask through a pixel-wise comparison of the input image and corresponding normal background, which requires a high-quality background reconstruction based on the input image (Cui et al., 2022). GANs have the advantage of generating realistic images from the learned manifold with sharp and clear detail, which makes GAN-inversion a promising tool for background reconstruction in pixel-wise anomaly detection. GAN-inversion for unsupervised anomaly detection (Xia et al., 2022b) was first introduced by (Schlegl et al., 2017) and various followup works have been proposed (Zenati et al., 2018; Schlegl et al., 2019; Baur et al., 2018; Kimura et al., 2020; Akcay et al., 2018). Counter-intuitively, instead of pixel-wise anomaly detection, the applications of GAN-based anomaly detection methods mainly focus on image-level/localization level (Xia et al., 2022a) with less satisfactory performance. For example, as one of the benchmark methods on the MVTec dataset (Bergmann et al., 2019), AnoGAN (Schlegl et al., 2017) performs the worst on image segmentation (even localization) compared to vanilla AE, not to mention the state-of-the-art methods (Yu et al., 2021; Roth et al., 2022). This is due to two intrinsic issues of GAN-inversion under unknown gross corruptions: (i) Lack of robustness: due to the existence of the anomalous region, the reconstructed normal background can easily deviate from the ground truth background (Figure 1); (ii) Gap between the approximated and actual manifolds (Pan et al., 2021): even for a clean input image, it is difficult to identify a latent representation that can achieve perfect reconstruction. When the residual is used for pixel-wise anomaly detection, those two issues will easily deteriorate its performance.

---

[1]Note that the "robustness to defects" mentioned in (Abdal et al., 2019) means that the image together with the defects can be faithfully restored in the latent space, instead of restoring a defect-free image

## 3 ROBUST GAN-INVERSION

In this section, we first give a problem overview. Then, we present the RGI method with a theoretical justification of its asymptotic robustness properties. A simulation study is conducted to verify the robustness. Next, we generalize the proposed method to R-RGI to mitigate the GAN approximation gap. Finally, we give a discussion that connects the proposed method with existing methods.

**Overview.** Given a pre-trained GAN network on a large-scale clean image dataset, such that the generator learns the image manifold. For any input image from the same manifold with unknown gross corruptions, we aim to restore a clean image and a corrupted region mask (Figure 2).

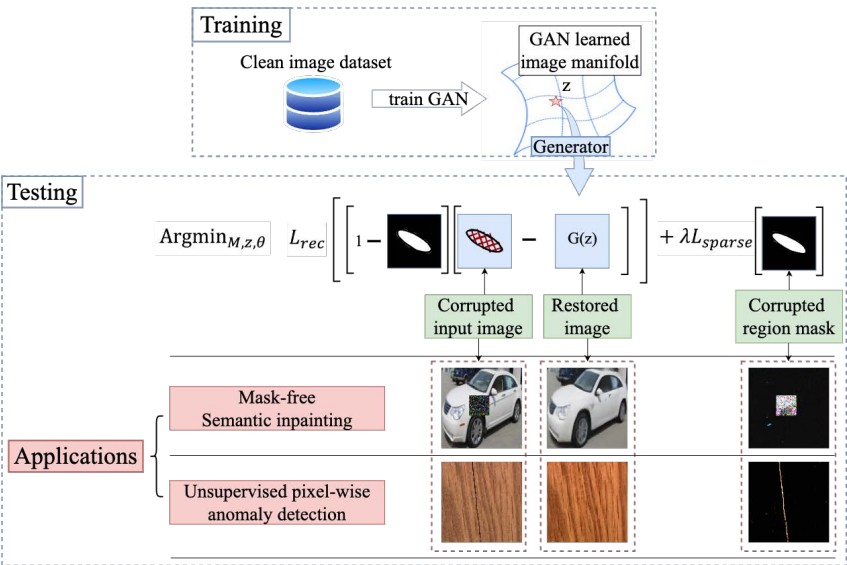

Figure 2: RGI/R-RGI for mask semantic inpainting and unsupervised anomaly detection: Given a pre-trained GAN network on a large-scale clean image dataset, such that the generator learns the image manifold. For any input image from the same manifold with unknown gross corruptions, the proposed RGI/R-RGI can restore a clean image $G(x)$ and and identify the corrupted region mask $M$ by solving the optimization problems thereof. Therefore, we unifies two important applications: (i) mask-free semantic inpainting; and (ii) unsupervised pixel-wise anomaly detection

**Notation**. Before we introduce the RGI method, we first introduce the following notations: For any positive integer $k$, we use $[k]$ to denote the set $\{1, 2, \ldots, k\}$; For any index set $\Lambda \subseteq [m] \times [n]$, we use $|\Lambda|$ to denote the cardinality of $\Lambda$; For any matrix $T$, the $l$-norm $\|T\|_l$ (e.g. $\|T\|_1$, $\|T\|_\infty$) is calculated by treating $T$ as a vector, and we use $I_T$ to denote the non-zero mask of $T$, i.e. $(I_T)_{ij} = 0$ if $T_{ij} = 0$ and 1 otherwise; For any two sets $A$ and $B$, we use $d_l^H(A, B) := \sup_{a \in A} \inf_{b \in B} \|a - b\|_l$ to denote the one-sided $l$-norm Hausdorff distance between $A$ and $B$, noting that when $A$ is a singleton, it becomes the standard $l$-norm distance $d_l(a, B) := \inf_{b \in B} \|a - b\|_l$; For any two matrices $A$ and $B$, we use $\odot$ to denote element-wise product, i.e., $(A \odot B)_{ij} = A_{ij} B_{ij}$.

### 3.1 ROBUST GAN-INVERSION

Assume that the GAN learns an accurate image manifold, i.e., there is no approximation gap between the GAN learned image manifold and true image manifold, such that any input image $x \in \mathbb{R}^{m \times n}$ with gross corruptions $s^* \in \mathbb{R}^{m \times n}$ follows: $x = G(z^*) + s^*$, where $z^* \in \mathbb{R}^d$ is the true latent code and $G(\cdot)$ is a pre-trained generator, i.e., $G(\cdot) : \mathbb{R}^d \to \mathbb{R}^{m \times n}$. Further, assume that $s^*$ admits sparsity property, i.e., $\|s^*\|_0 \leq n_0$, where $n_0$ is the number of corrupted pixels. Given $x$, we aim to restore $G(z^*)$ (or $z^*$), and consequently, achieve (i) semantic image inpainting, i.e., $G(z^*)$, or (ii) pixel-wise anomaly detection, i.e., $M^* = I_{x - G(z^*)}$. To achieve so, we propose to learn the latent representation $z$ and the corrupted region mask $M$ at the same time, i.e.,

$$\min_{z \in \mathbb{R}^d, M \in \mathbb{R}^{m \times n}} L_{rec}((\mathbf{1} - M) \odot x, (\mathbf{1} - M) \odot G(z)) \qquad \text{s.t.} \quad \|M\|_0 \leq n_0. \qquad (2)$$

The reconstruction loss term $L_{rec}(\cdot, \cdot)$ measures the distance between the input image and the generated one outside the corrupted region, which guides the optimization process to find the latent variable $z$. Intuitively, when solving for the mask along with the latent variable, we aim to allocate the $n_0$ mask elements such that reconstruction loss is minimized. It is easy to check that the true solution $G(z^*)$ and $M^*$ is optimal. Moreover, if we assume that $z^*$ is the only latent code such that $\|x - G(z^*)\|_0 \leq n_0$, then we have uniqueness.

However, Equation 2 with $\|\cdot\|_0$ is hard to solve. To address this issue, we relax Equation 2 to an unconstrained optimization problem that can be solved directly using gradient decent algorithms:

$$\min_{z \in \mathbb{R}^d, M \in \mathbb{R}^{m \times n}} L_{rec}((\mathbf{1} - M) \odot x, (\mathbf{1} - M) \odot G(z)) + \lambda \|M\|_1. \tag{3}$$

Equation 3 is named **RGI**, where the second term penalizes the mask size to avoid a trivial solution with the mask expanding to the whole image. Intuitively, the second term encourages a small mask; however, the reconstruction loss will increase sharply once the learned mask cannot cover the corrupted region. By carefully selecting the tuning parameter $\lambda$, we will arrive at a solution with (i) a high-quality image restoration with negligible reconstruction error; (ii) an accurate mask that covers the corrupted region. The following two theorems justify our intuition.

**Theorem 1** *(Asymptotic optimality of $z$) Assume (i) GAN learns an accurate image manifold, i.e., there exists $z^*$ such that $\|x - G(z^*)\|_0 \leq n_0$; (ii) $z$ is bounded for both Equation 2 and Equation 3, or equivalently there exists $R > 0$ such that $\|z\|_1 \leq R$, i.e., $z \in \mathbb{S}^d$ with $\mathbb{S}^d := [-R, R]^d$; (iii) $L_{rec}(\cdot) = \|\cdot\|_2^2$; and (iv) $G(z)$ is continuous. Let $\hat{z}(\lambda)$ be any optimal $z$ solution of Equation 3 with tuning parameter $\lambda$, and $\mathcal{Z}^*$ be the optimal $z$ solution set of Equation 2, we have $d_\infty(\hat{z}(\lambda), \mathcal{Z}^*) \downarrow 0$ as $\lambda \downarrow 0$. Moreover, denote $\tilde{n} = \min_{z \in \mathbb{S}^d} \|x - G(z)\|_0$ and $\tilde{\mathcal{Z}} = \{z \in \mathbb{S}^d \mid \|x - G(z)\|_0 = \tilde{n}\}$, we have $d_\infty(\hat{z}(\lambda), \tilde{\mathcal{Z}}) \downarrow 0$ as $\lambda \downarrow 0$. If we further assume a unique $z^* = \arg\min_{z \in \mathbb{S}^d} \|x - G(z)\|_0$, i.e., $\tilde{\mathcal{Z}} = \{z^*\}$, then $\hat{z}(\lambda) \to z^*$ as $\lambda \downarrow 0$.*

Note that Assumption (ii) is only for the proof purpose. We could always choose a large enough $R$ to include all possible optimal solutions so that the optimality of Equation 2 and Equation 3 remains. **Remark**: Theorem 1 states that the optimal $z$ solution of the RGI method, $G(\hat{z})$, converges to the true background $G(z^*)$ as $\lambda \downarrow 0$, regardless of the corruption magnitude, which proves the robustness of the RGI method to unknown gross corruptions, and is the key to image restoration.

**Theorem 2** *(Asymptotic optimality of $M$) Follow the same assumptions and notations in Theorem 1. Let $\hat{M}(\lambda)$ be any optimal $M$ solution of (3), and $\tilde{\mathcal{M}} := \{I_{(x - G(\tilde{z}))} | \tilde{z} \in \tilde{Z}\} \subseteq \{M \in \{0, 1\}^{m \times n} | \|M\|_0 \leq \tilde{n}\}$. We have $d_\infty(\hat{M}(\lambda), \tilde{\mathcal{M}}) \downarrow 0$ as $\lambda \downarrow 0$. Moreover, there is a finite $\tilde{\lambda} > 0$ such that for any $\lambda \leq \tilde{\lambda}$, there is an $\tilde{M} \in \tilde{\mathcal{M}}$ such that $\tilde{M} = I_{\hat{M}(\lambda)}$. If we further assume $\tilde{\mathcal{Z}} = \{z^*\}$, then (i) $\hat{M}(\lambda) \to M^*$ as $\lambda \downarrow 0$, and (ii) for any $\lambda \leq \tilde{\lambda}$, $I_{\hat{M}(\lambda)} = M^*$.*

**Remark**: Theorem 2 states that the optimal $M$ solution of the RGI method, $\hat{M}$, converges to the true corrupted region mask $M^*$ as $\lambda \downarrow 0$, regardless of the corruption magnitude. Moreover, there is a fixed $\tilde{\lambda}$, if we choose a tuning parameter $\lambda \leq \tilde{\lambda}$, the true corrupted regions mask can be identified by simply thresholding the $\hat{M}$, i.e., $M^* = I_{\hat{M}(\lambda)}$, which is the key for pixel-wise anomaly detection. The proof of Theorems 1 and 2 are provided in Appendix B. A simulation study to verify the robustness of proposed RGI method is provided in Appendix C.

## 3.2 RELAXED ROBUST GAN-INVERSION

In traditional GAN-inversion methods (Yeh et al., 2017; Pan et al., 2021), without mask information, fine-tuning the generator parameters will lead to severe overfitting towards the input image. However, fine-tuning is the key step to mitigate the gap between the learned image manifold and any specific input image (Pan et al., 2021). The proposed approach makes fine-tuning possible, i.e.,

$$\min_{z \in \mathbb{R}^d, M \in \mathbb{R}^{m \times n}, \theta \in \mathbb{R}^w} L_{rec}((\mathbf{1} - M) \odot x, (\mathbf{1} - M) \odot G(z; \theta)) + \lambda \|M\|_1. \tag{4}$$

Equation 4 is named **R-RGI**. This problem can also be solved directly using gradient decent types of algorithm with carefully designed parameters for learning parameter $\theta$. We found the following

strategy gives better performance: at the beginning of the solution process, we fix $\theta$. When we get a stable reconstructed image as well as the mask, then we optimize $\theta$ together with all the other decision variables with a small step size for limited iterations.

In this section, we address the robustness of GAN-inversion methods and show the asymptotic optimality of the RGI method. Moreover, the R-RGI enables fine-tuning of the learned manifold towards a specific image for better restoration quality and thus improves the performance of both tasks.

### 3.3 DISCUSSIONS

**Connection to robust machine learning methods**. The RGI method roots in robust learning methods (Caramanis et al., 2012; Gabrel et al., 2014), which aims to restore a clean signal (or achieve robust parameter estimation) in the presence of corrupted input data. Robust machine learning methods, including robust dimensionality reduction (Candès et al., 2011; Xu et al., 2010; Peng et al., 2012), matrix completion (Candès & Recht, 2009; Jain et al., 2013), use statistical priors to model the signal to be restored, such as low rank, total variation, etc. Those statistical priors limit its applications involving complex natural images, e.g., the restoration of corrupted human face images.

Similarly, we also aim for signal restoration from a corrupted input signal, but with two key differences: (i) instead of the restrictive statistical priors, we adopt a learned deep generative prior (Pan et al., 2021), i.e., $G(z)$, which plays a key role in modeling complex natural images. (ii) instead of recovering the corruptions, we learn a sparse binary mask $M$ that covers the corrupted region, which is much easier than learning corruptions itself. The RGI method significantly extends the traditional robust machine learning methods to a wider range of applications.

**Connection to robust statistics**. The proposed method also has a deep connection with traditional robust statistics (Huber, 2011): when adopting an $l_2$-norm reconstruction loss, the loss function of Equation 3 can be simplified as $\sum_{ij} f_{ij}(z;\lambda)$ where $f_{ij}(z;\lambda) =$
$$\begin{cases} (x - G(z))_{ij}^2, & \text{if } 2(x - G(z))_{ij}^2 < \lambda \\ \lambda - \frac{\lambda^2}{4(x-G(z))_{ij}^2}, & \text{otherwise} \end{cases}$$ (Equation 6 in the proof of Theorem 1), which shares a similar spirit as $M$-estimators, e.g., metric Winsorizing and Tukey's biweight, thus inherits the robustness with respect to outliers. Moreover, Equation 3 allows a flexible way of incorporating robustness to reconstruction loss functions beyond convex formulations, such as the perceptual loss and discriminator loss (Pan et al., 2021).

## 4 CASE STUDY

### 4.1 MASK-FREE SEMANTIC INPAINTING

Semantic inpainting is an important task in image editing and restoration. (Please see Appendix A.1 for a comprehensive literature review on this topic.) Among all the methods, GAN-inversion based methods have the advantage of inpainting a single image with arbitrary missing regions without any requirement for mask information in the training stage. However, the requirement of a pre-configured corrupted region mask during testing hinders its application in mask-free semantic inpainting. In this section, we aim to show that the RGI method can achieve mask-free semantic inpainting by inheriting the mask-free training nature of GAN-inversion based methods, while avoiding the pre-configured mask requirement during testing. Therefore, we will compare with the state-of-the-art GAN-inversion based image inpainting methods that projecting onto the $\mathcal{Z}$ space, including (a) Yeh et al. (2017) without a pre-configured mask (Yeh et al. (2017) w/o mask) baseline; (b) Yeh et al. (2017) with a pre-configured mask ((Yeh et al., 2017) w/ mask); and (c) Pan et al. (2021) with a pre-configured mask (Pan et al. (2021) w/ mask).

**Datasets and metrics**. We evaluate the proposed methods on three datasets, the CelebA (Liu et al., 2015), Stanford car (Krause et al., 2013), and LSUN bedroom (Yu et al., 2015), which are commonly used for benchmarking image editing algorithms. We consider two different cases: (i) central block missing, (ii) random missing and (iii) irregular shape missing (see Appendix I). We fill in the missing entry with pixels from $N(-1, 1)$. PSNR and SSIM are used for performance evaluation. Please see Appendix D for implementation details.

Table 1: Semantic inpainting performance of Yeh et al. (2017) (w/ and w/o mask), Pan et al. (2021) (w/ mask), RGI and R-RGI

| Datasets | Cases | metrics | methods | | | | |
|---|---|---|---|---|---|---|---|
| | | | Yeh et al. w/o mask | Yeh et al. w/ mask | **RGI** | Pan et al. w/ mask | **R-RGI** |
| CelebA | Case (i) | PSNR ↑ | 11.50 | 20.82 | **19.70** | 21.74 | **20.05** |
| | | SSIM ↑ | 0.358 | 0.492 | **0.451** | 0.570 | **0.509** |
| | Case (ii) | PSNR ↑ | 19.64 | 22.63 | **21.52** | 27.63 | **23.73** |
| | | SSIM ↑ | 0.440 | 0.536 | **0.490** | 0.766 | **0.655** |
| Cars | Case (i) | PSNR ↑ | 16.57 | 17.50 | **16.89** | 20.98 | **19.31** |
| | | SSIM ↑ | 0.359 | 0.377 | **0.363** | 0.636 | **0.618** |
| | Case (ii) | PSNR ↑ | 17.36 | 17.71 | **17.52** | 21.61 | **21.18** |
| | | SSIM ↑ | 0.361 | 0.382 | **0.363** | 0.650 | **0.588** |
| LSUN bedroom | Case (i) | PSNR ↑ | 16.15 | 19.27 | **17.67** | 21.36 | **18.72** |
| | | SSIM ↑ | 0.405 | 0.428 | **0.416** | 0.587 | **0.567** |
| | Case (ii) | PSNR ↑ | 19.26 | 19.66 | **19.72** | 22.30 | **22.29** |
| | | SSIM ↑ | 0.419 | 0.433 | **0.420** | 0.599 | **0.557** |

**Comparison Results.** The PSNR and SSIM of image restoration are shown in Table 1. We can observe that (i) the RGI outperforms the Yeh et al. (2017) w/o mask baseline, and achieves a comparable performance with Yeh et al. (2017) w/ mask – the best possible result without fine-tuning the generator. However, there is no pre-configured mask requirement in the RGI method, which demonstrates RGI's robustness to unknown gross corruptions. Such performance improvement is significant, especially on CelebA dataset, where GAN learns a high quality face manifold (high SSIM/PSNR value in Yeh et al. (2017) w/ mask). (ii) the R-RGI further improves the image restoration performance with fine-tuning the generator, which achieves a comparable performance with the (Pan et al., 2021) w/ mask – the best possible result with fine-tuning the generator. Such performance improvement is significant, especially on Stanford cars and LSUN bedroom datasets, where even the performance of Yeh et al. (2017) w/ mask is limited, indicating a large GAN approximation gap. As shown in Figure 3, the mask-free generator fine-tuning by R-RGI guarantees a high-quality image restoration. More qualitative results are in Appendix D.

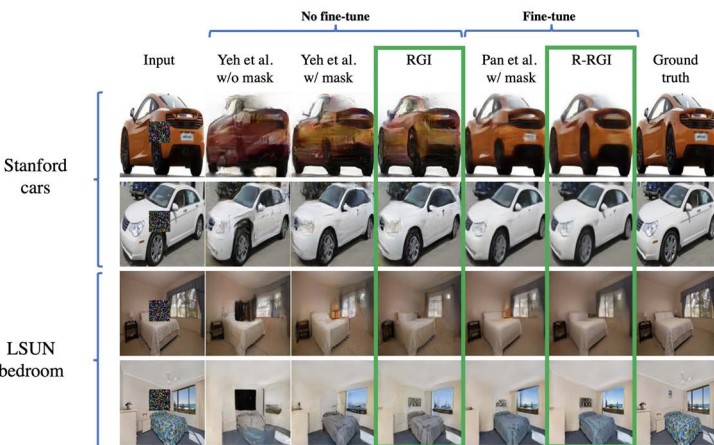

Figure 3: Case (i): Restored images on Cars (Krause et al., 2013) and LSUN (Yu et al., 2015).

## 4.2 UNSUPERVISED PIXEL-WISE ANOMALY DETECTION

Unsupervised pixel-wise anomaly detection is becoming important in product cosmetic inspection. The extracted pixel-wise accurate defective region masks are then used for various downstream tasks, including aiding pixel-wise annotation, providing precise defect specifications (i.e. diameter,

area) for product surface quality screening, and so on, which cannot be achieved by current sample level/localization level algorithms. The RGI/R-RGI method is developed for such an unsupervised fine-grained surface quality inspection task in a data-rich but defect/annotation-rare environment, which is common for mass production such as consumer electronics, steel manufacturing, and so on. In those applications, it is cheap to collect a large number of defect-free product images, while expensive and time-consuming to collect and annotate defective samples due to the super-high yield rate and expert annotation requirements.

There are three categories of unsupervised pixel-wise anomaly detection methods, including robust optimization based methods, deep reconstruction based methods (including GAN-inversion based methods), and deep representation based methods (Cui et al., 2022) (Please see Appendix A.2 for a comprehensive literature review). In addition to the AnoGAN (Schlegl et al., 2017) baseline, we will compare with the SOTA method in each category, including the RASL (Peng et al., 2012), the SOTA method in robust optimization method, which improves the RPCA (Candès et al., 2011) to solve the linear misalignment issue; DRAEM (Zavrtanik et al., 2021) which is the SOTA method in deep-reconstruction based methods; and PatchCore (Roth et al., 2022), a representative deep representation based method that performs the best on the MVTec (Bergmann et al., 2019) dataset. We aim to show that with a simple robustness modification, the RGI/R-RGI will significantly improve the baseline AnoGAN's performance and outperform the SOTA. We use a PGGAN (Karras et al., 2017) as the backbone network and a $l_2$ norm reconstruction loss term ($L_{rec}$). For AnoGAN, the pixel-wise reconstruction residual $|\hat{x}-x|$ is used as the defective region indicator. We apply a simple thresholding of the residual and report the Dice coefficient of the best performing threshold.

**Datasets.** We notice that the popular benchmark datasets, including MVTec AD (Bergmann et al., 2019) and BTAD (Mishra et al., 2021) for industrial anomaly detection, are not suitable for this task due to the following reasons: (i) the annotation of those datasets tends to cover extra regions of the real anomaly contour, which favors localization level methods. (ii) The number of clean images in most of the categories is small (usually $200 \sim 300$ images), which may not be sufficient for GAN training. A detailed discussion of the MVTec dataset can be found in Appendix E. To gain a better control of the defect annotation and better reflect the data-rich but defect/annotation-rare application scenario, we generate a synthetic defect dataset based on Product03 from the BTAD (Mishra et al., 2021) dataset. The synthetic dataset contains 900 defect-free images for training and 4 types of defects for testing, including crack, scratch, irregular, and mixed large (100 images in each category). Qualitative and quantitative comparisons with SOTA methods will be conducted on this dataset. The synthetic defect generation process are provided in Appendix F.

**Metrics.** We use the Dice coefficient to evaluate the pixel-wise anomaly detection performance, which is widely adopted for image segmentation tasks. Dice coefficient is defined as $(2\|\hat{M} \odot M\|_0)/(\|\hat{M}\|_0 + \|M\|_0)$, where $\hat{M} \in \mathbb{R}^{m \times n}$ is the predicted binary segmentation mask for the anomalous region and $M \in \mathbb{R}^{m \times n}$ is the true binary segmentation mask with 1 indicating the defective pixels and 0 otherwise. Notice that pixel-wise AUROC score (Bergmann et al., 2019) is sensitive to class imbalance, which may give misleading results in defective region segmentation tasks when the defective region only covers a small portion of pixels in the whole image (This is often the case in industrial cosmetic inspection or medical applications (Baur et al., 2021; Mou et al., 2022; Zavrtanik et al., 2021). We mainly compare the Dice coefficients for different methods.

**Compare with the AnoGAN baseline on on synthetic defect dataset.** The results are shown in Table 2 and Figure 4 (a). (i) Compared to AnoGAN(Schlegl et al., 2017), the only modification in the RGI method is the additional sparsity penalty term of the anomalous region mask $M$ to enhance its robustness. However, with such a simple modification, RGI significantly outperforms the AnoGAN's performance under large defects ('mix large'), where the large anomalous region can easily lead to a deviation between the AnoGAN restored image and the real background. (ii) The R-RGI achieves a significant and consistent performance improvement over the RGI and AnoGAN methods. The generator fine-tuning process closes the gap between the GAN learned normal background manifold and the specific test image, which leads to better background restoration and mask refinement. The implementation details and more qualitative results can be found in Appendix G.

**Compare with the SOTA methods on on synthetic defect dataset**. The results are shown in Table 2 and Figure 4 (b). R-RGI method performs the best in all defect types. The limited modeling capability of the low-rank prior used in RASL (Peng et al., 2012) leads to its bad performance; As a localization level method, PatchCore (Roth et al., 2022) can successfully localize the defect (Figure

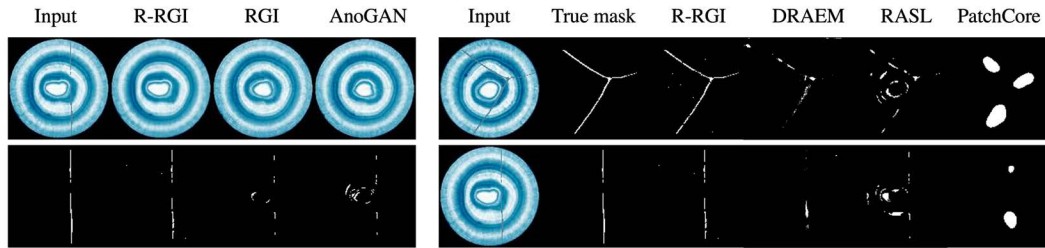

(a) Comparison Among R-RGI, RGI and AnoGAN: The first row: input image and restored backgrounds; The second row: true mask and detected defect mask.

(b) Comparison Among R-RGI, DRAEM, RASL and PatchCore: The first column indicates the input images, and the following columns are detected defect mask.

Figure 4: Performance comparison among different methods.

Table 2: Pixel-wise anomaly detection performance on synthetic defect dataset

| methods | metrics | Defects | | | |
|---|---|---|---|---|---|
| | | crack | irregular | scratch | mixed large |
| RASL | Dice ↑ | 0.293 (0.101) | 0.246 (0.181) | 0.280 (0.167) | 0.355 (0.100) |
| | AUROC ↑ | 0.868 (0.044) | 0.817 (0.061) | 0.834 (0.079) | 0.784 (0.066) |
| PatchCore | Dice ↑ | 0.277 (0.157) | 0.448 (0.167) | 0.350 (0.103) | 0.683 (0.138) |
| | AUROC ↑ | 0.936 (0.040) | 0.973 (0.036) | 0.966 (0.033) | 0.972 (0.029) |
| AnoGAN | Dice ↑ | 0.457 (0.117) | 0.422 (0.208) | 0.436 (0.212) | 0.540 (0.131) |
| | AUROC ↑ | 0.954 (0.021) | 0.956 (0.030) | 0.947 (0.030) | 0.949 (0.023) |
| RGI | Dice ↑ | **0.458 (0.119)** | **0.420 (0.194)** | **0.430 (0.192)** | **0.617 (0.110)** |
| | AUROC ↑ | **0.951 (0.021)** | **0.952 (0.027)** | **0.944 (0.029)** | **0.949 (0.022)** |
| DRAEM | Dice ↑ | 0.473 (0.212) | 0.697 (0.152) | 0.639 (0.169) | 0.825 (0.108) |
| | AUROC ↑ | 0.951 (0.035) | 0.988 (0.013) | 0.982 (0.017) | 0.989 (0.195) |
| R-RGI | Dice ↑ | **0.809 (0.109)** | **0.758 (0.135)** | **0.745 (0.167)** | **0.810 (0.064)** |
| | AUROC ↑ | **0.977 (0.037)** | **0.988(0.015)** | **0.980 (0.020)** | **0.968 (0.015)** |

(Roth et al., 2022)). However, the loss of resolution deteriorates its pixel-level anomaly detection performance; The DRAEM (Zavrtanik et al., 2021) jointly trains a reconstructive sub-network and a discriminative sub-network with additional simulated anomaly samples on top of the clean training images. Its performance highly relies on the coverage of the simulated anomaly samples and is more sensitive to large anomalies. More importantly, by incorporating the so-called mask free fine-tuning, the R-RGI method successfully improves the baseline AnoGAN method's performance over those SOTA methods on this task. More qualitative results can be found in Appendix G.

## 5 CONCLUSION

Robustness has been a long pursuit in the field of signal processing. Recently, utilizing GAN-inversion for signal restoration gains popularity in various signal processing applications, since it demonstrates strong capacity in modeling the distribution of complex signals such as natural images. However, there is no robustness guarantee in the current GAN-inversion method.

To improve the robustness and accuracy of GAN-inversion in the presence of unknown *gross* corruptions, we propose an RGI method. Furthermore, we prove the asymptotic robustness of the proposed method, i.e., (i) the restored signal from RGI converges to the true clean signal (for image restoration); (ii) the identified mask converges to the true corrupted region mask (for anomaly detection). Moreover, we generalize RGI method to R-RGI method to close the GAN approximation gap, which further improves the image restoration and unsupervised anomaly detection performance.

The RGI/R-RGI method unifies two important tasks under the same framework and achieves SOTA performance: (i) Mask-free semantic inpainting, and (ii) Unsupervised pixel-wise anomaly detection.

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

## A COMPREHENSIVE LITERATURE REVIEW

This section provides comprehensive literature reviews of mask-free semantic inpainting and unsupervised pixel-wise anomaly detection, including but not limited to GAN-inversion based methods.

### A.1 COMPREHENSIVE LITERATURE REVIEW FOR MASK-FREE SEMANTIC INPAINTING

**Mask-free Semantic inpainting** aims to restore the corrupted region of an input image with little or no information on the corruptions. To achieve this goal, multiple traditional single image semantic inpainting methods exploit fixed image priors, including total variation (Afonso et al., 2010; Shen & Chan, 2002), low rank (Hu et al., 2012), patch off set statistics (He & Sun, 2012) and so on.

However, due to the fixed image prior, those method have strong assumptions on the input image, such as smoothness, containing similar structure or patches, which may fail when dealing with large missing regions with novel content, i.e., recovering the nose or mouth in facial images (Yeh et al., 2017).

Notice that various convolutional neural network based methods for semantic inpainting has been proposed (Pathak et al., 2016; Iizuka et al., 2017; Li et al., 2017; Yu et al., 2018; Liu et al., 2018; Yu et al., 2019; Li et al., 2020; Suvorov et al., 2022; Song et al., 2018; Yan et al., 2018; Liu et al., 2019a;b; Nazeri et al., 2019; Ren et al., 2019; Xiong et al., 2019; Zeng et al., 2019; 2020; Zhao et al., 2021; Zhu et al., 2021). In addition to the requirement of a pre-configured mask for inpainting an input image, they usually need mask information in the training stage, either the same fixed mask as the region to be inpainted (Pathak et al., 2016), or trying to cover irregular missing regions by randomly sampling a rectangular mask with random location (Yang et al., 2017), or a fixed set of irregular masks (Liu et al., 2018) or generating masks following a set of rules (Yu et al., 2019; Zhao et al., 2021). Those methods cannot fulfill the mask-free semantic inpainting goal.

Another closely related field is named blind image inpainting, which aims to inpaint a single corrupted image without the need for the corrupted regions mask (Liu et al., 2019b; Wang et al., 2020; Qian et al., 2018; Wu et al., 2019; El Helou & Süsstrunk, 2022). However, most of them need a training set of possible corruptions (and/or corrupted region masks), which again restricts their ability to generalize to unknown gross corruptions. Thus, they are not mask-free methods.

GAN-inversion for semantic inpainting was first introduced by Yeh et al. (2017) and was further developed by Gu et al. (2020); Pan et al. (2021); Wang et al. (2022b) for improving inpainting quality. They have the advantage of inpainting a single image with arbitrary missing regions, without any requirement for mask information in the training stage. However, they do require a pre-configured corrupted region mask for reliable inpainting during testing.

The RGI method inherits the mask-free training nature of GAN-inversion based semantic inpainting methods, while avoiding the pre-configured mask requirement during testing. Thus, we can achieve mask-free semantic inpainting for a single test image with arbitrary gross corruptions.

## A.2 COMPREHENSIVE LITERATURE REVIEW FOR UNSUPERVISED PIXEL-WISE ANOMALY DETECTION

**Unsupervised pixel-wise anomaly detection** aims to extract a pixel-level segmentation mask for anomalous regions, which plays an important role in industrial and medical applications (Yan et al., 2017; Baur et al., 2021). Unlike image-level (identify anomalous samples) or localization level (i.e., localize anomaly) anomaly detection, unsupervised pixel-wise anomaly detection extracts the anomalous region segmentation mask through a pixel-wise comparison of the input image and corresponding normal background. Therefore, it requires a high-quality background reconstruction based on the input image. To achieve this goal, robust optimization methods rely on the statistical prior knowledge of the background (such as low-rank (Bouwmans & Zahzah, 2014) and smoothness (Yan et al., 2017)), which is effective when the true background satisfies those assumptions. However, such assumptions can be restrictive and highly dependent on the properties of background image for specific applications. In contrast, the deep reconstruction based methods (Pang et al., 2021) methods reconstruct the normal background from a learned subspace and assume such a subspace does not generalize to anomalies. Autoencoder (AE) (Bergmann et al., 2018), variational AE (VAE) (Kingma & Welling, 2013) and its variants (please see the review paper (Baur et al., 2021)) are popular tools. However, such assumptions may not always hold, i.e., an AE that achieves a satisfactory reconstruction of normal regions of the input image also "generalize" so well that it can always reconstruct the abnormal inputs as well (Gong et al., 2019). Some solutions, such as MemAE (Gong et al., 2019) and PAEDID (Mou et al., 2022) restrict this generation capability by reconstructing the background from a memory bank of clean training images, DRAEM Zavrtanik et al. (2021) restrict this generation capability by integrating an discriminated network. Another category of unsupervised approaches are deep representation-based methods, which learns the discriminate embeddings of normal images from a clean training set and achieve anomaly detection by comparing the embedding of a test image and the distribution of the normal image embeddings, such as PatchCore(Roth et al., 2022), Padim (Defard et al., 2021), Cflow (Gudovskiy et al., 2022) STFPM (Wang et al.,

2021). Those methods usually serves as localization tools since the comparison in the embedded space will lead to the loss of resolution.

Recently GAN-based anomaly detection methods have gained popularity in reconstruction based methods (Xia et al., 2022b). GANs have the advantage of generating realistic images from the learned manifold with sharp and clear detail, regardless of image type (Pang et al., 2021), which make GAN a promising tool for background reconstruction for pixel-wise anomaly detection. Inspired by this idea, Schlegl et al. (2017) borrowed the GAN-inversion for unsupervised anomaly detectionand various followup works has been proposed, including EBGAN (Zenati et al., 2018), f-AnoGAN (Schlegl et al., 2019) and GANomaly (Akcay et al., 2018), which mainly focus on improving inference speed. Counterintuitively, instead of pixel-wise anomaly detection, the applications of GAN-based anomaly detection methods mainly focus on image-level/localization level (Xia et al., 2022a) with a less satisfactory performance. For example, as one of the benchmark methods on the MVTec dataset (Bergmann et al., 2019), AnoGAN (Schlegl et al., 2017) performs the worst on image segmentation (even localization) compared to vanilla AE, not to say the state-of-the-art methods (Yu et al., 2021; Roth et al., 2022). This is due to the intrinsic issues of GAN-inversion: (i) Lack of robustness: due to the existence of the anomalous region, the reconstructed normal background can easily deviate from the ground truth background (Figure 1); (ii) Gap between the approximated and actual manifolds (Pan et al., 2021): even for a clean input image, it is difficult to identify a latent representation that can achieve perfect reconstruction. When the residual is used for pixel-wise anomaly detection, those issues will easily deteriorate its performance.

We aim to demonstrate the performance improvement of GAN-based anomaly detection method by RGI, which makes the GAN-inversion-based methods practical in pixel-wise anomaly detection tasks.

## B   PROOF TO THEOREMS 1 AND 2

### B.1   PROOF TO THEOREMS 1

Under the assumption, there exists $z^*$ such that $\|x - G(z)\|_0 \leq n_0$. Thus $(z^*, M^*)$ solves Equation 2 to its optimal value of 0.

Denote $\mathcal{Z}^{**} = \{z \in \mathbb{S}^d \mid \|x - G(z)\| \leq n_0\}$. Note that for any $z \in \mathcal{Z}^{**}$, we could set $M = I_{x-G(z)}$ with $\|M\|_0 \leq n_0$ and then $(z, M)$ also solves Equation 2 to its optimal value of 0. On the other hand, for any $z \notin \mathcal{Z}^{**}$, i.e., $\|x - G(z)\|_0 \geq n_0 + 1$, $\|(1 - M) \odot (x - G(z))\|_2^2 > 0$ unless $\|M\| \geq \|x - G(z)\|_0 \geq n_0 + 1$, which renders such $M$ is infeasible. Thus, we conclude that $\mathcal{Z}^{**} = \mathcal{Z}^*$.

The same optimality arguments apply to every $\tilde{z} \in \tilde{\mathcal{Z}}$ as $\tilde{\mathcal{Z}} \subseteq \mathcal{Z}^*$. We next prove that Equation 3 asymptotically converges to $\tilde{\mathcal{Z}}$, which will complete the proof.

Denote $f(z, M; \lambda) := \|(1 - M) \odot (x - G(z))\|_2^2 + \lambda \|M\|_1$ the objective function of Equation 3. Select any $\tilde{z} \in \tilde{\mathcal{Z}}$ and let $\tilde{M} = I_{x-G(\tilde{z})}$, and we note that $f(\tilde{z}, \tilde{M}; \lambda) = \lambda \tilde{n}$.

Now consider for any given $z$, we could next calculate $\hat{M}(z)$ which minimizes $f(z, \hat{M}; \lambda)$. Note that

$$f(z, M; \lambda) = \sum_{i,j} ((1 - M_{ij})(x - G(z))_{ij}^2 + \lambda |M_{ij}|) := \sum_{i,j} f_{ij}(z, M_{ij}; \lambda)$$

with

$$\frac{\partial f}{\partial M_{ij}} = \frac{\partial f_{ij}}{\partial M_{ij}} = 2(x - G(z))_{ij}^2 (M_{ij} - 1) + \lambda \partial |M_{ij}|,$$

where $\partial |M_{ij}|$ is the partial differential of $|M_{ij}|$.

It is clear that $\frac{\partial f_{ij}}{\partial M_{ij}} < 0$ for $M_{ij} < 0$ and $\frac{\partial f_{ij}}{\partial M_{ij}} > 1$ for $M_{ij} > 1$, and thus the optimal $\hat{M}_{ij} \in [0, 1]$. Within the interval $(0, 1)$, we in addition have

$$\frac{\partial f_{ij}}{\partial M_{ij}} = 2(x - G(z))_{ij}^2 (M_{ij} - 1) + \lambda, \tag{5}$$

and $\frac{\partial f_{ij}}{\partial M_{ij}} = 0$ solves to

$$\hat{M}_{ij}^* = 1 - \frac{\lambda}{2(x - G(z))_{ij}^2}.$$

Discussion:

- If $2(x - G(z))_{ij}^2 \geq \lambda$: $\frac{\partial f_{ij}}{\partial M_{ij}} < 0$ for $M_{ij} \in (0, \hat{M}_{ij}^*)$ and $\frac{\partial f_{ij}}{\partial M_{ij}} > 0$ for $M_{ij} \in (\hat{M}_{ij}^*, 1]$, proving the optimality of $\hat{M}_{ij} = \hat{M}_{ij}^*$.

- If $2(x - G(z))_{ij}^2 < \lambda$: $\frac{\partial f_{ij}}{\partial M_{ij}} > 0$, for $M_{ij} \in (0, 1)$, thus pointing to the optimal $\hat{M}_{ij} = 0$.

Combining those two cases, introduce $(\cdot)_+ := \max\{\cdot, 0\}$ and we have

$$\hat{M}_{ij} = \left(1 - \frac{\lambda}{2(x - G(z))_{ij}^2}\right)_+.$$

We now take the optimal $\hat{M}$ back to $f$ and get

$$f_{ij}(z, \hat{M}; \lambda) = \begin{cases} (x - G(z))_{ij}^2, & \text{if } 2(x - G(z))_{ij}^2 < \lambda, \\ \lambda - \frac{\lambda^2}{4(x - G(z))_{ij}^2}, & \text{otherwise.} \end{cases} \tag{6}$$

Define

$$\mu(\lambda) := \frac{\tilde{n} + 2}{4}\lambda \geq \lambda/2,$$

while also noting that $\mu \downarrow 0$ as $\lambda \downarrow 0$. For any $z \in \mathbb{S}^d$, define the index set

$$\Lambda_\lambda(z) = \{(i, j) \mid (x - G(z))_{ij}^2 > \mu(\lambda)\}$$

and

$$\mathcal{Z}(\lambda) = \{z \in \mathcal{S}^d \mid |\Lambda_\lambda(z)| \leq \tilde{n}\}.$$

We next show that any $z \notin \mathcal{Z}(\lambda)$ cannot be optimal. For any $z \notin \mathcal{Z}(\lambda)$, we have $|\Lambda_\lambda(z)| \geq \tilde{n} + 1$. We also note that when $(x - G(z))_{ij}^2 \geq \mu(\lambda) \geq \lambda/2$, we have

$$\hat{f}_{ij}(z; \lambda) > \lambda - \frac{\lambda^2}{4\mu(\lambda)} = (1 - \frac{1}{\tilde{n} + 2})\lambda.$$

Therefore

$$f(z, M; \lambda) > (\tilde{n} + 1)(1 - \frac{1}{\tilde{n} + 2})\lambda > \tilde{n}\lambda = f(\tilde{z}, \tilde{M}; \lambda),$$

proving the non-optimality of such $z$.

As we limit the optimal solution to $\mathcal{Z}(\lambda)$, we now show that $d_\infty^H(\mathcal{Z}(\lambda), \tilde{\mathcal{Z}}) \downarrow 0$ as $\lambda \downarrow 0$.

Assume the statement is not true, i.e., there exists $\epsilon_0 > 0$ such that $d_\infty^H(\mathcal{Z}(\lambda), \tilde{\mathcal{Z}}) \geq \epsilon_0$ for any $\lambda > 0$. Denote $\Xi = \{\Lambda \subseteq [m] \times [n] \mid |\Lambda| \leq \tilde{n}\}$, which is clearly finite. For any $\Lambda \in \Xi$, denote

$$\mathcal{Z}^\Lambda(\lambda) = \{z \in \mathbb{S}^d \mid (x - G(z))_{ij}^2 \leq \mu(\lambda), \forall (i, j) \in \Lambda\}$$

and we have the following decomposition of $\mathcal{Z}(\lambda)$:

$$\mathcal{Z}(\lambda) = \cup_{\Lambda \in \Xi} \mathcal{Z}^\Lambda(\lambda),$$

which is mathematically saying that $\mathcal{Z}(\lambda)$ can be decomposed by enumerating all the possible cases of choosing $\tilde{n}$ elements from $[m] \times [n]$. Combined with the fact that $\Xi$ is finite, we have $d_\infty^H(\mathcal{Z}(\lambda), \tilde{\mathcal{Z}}) = \max_{\Lambda \in \Xi} d_\infty^H(\mathcal{Z}^\Lambda(\lambda), \tilde{\mathcal{Z}}) \geq \epsilon_0$.

Note that as $\mathcal{Z}(\lambda)$ decreasing with respect to $\lambda$, $d_\infty^H(\mathcal{Z}(\lambda), \tilde{\mathcal{Z}})$ is also decreasing, and the same applies to $\mathcal{Z}^\Lambda(\lambda)$ for any $\Lambda \in \Xi$. Therefore, there exists a particular $\Psi \in \Xi$ such that $d_\infty^H(\mathcal{Z}^\Psi(\lambda), \tilde{\mathcal{Z}}) \geq \epsilon_0$ for any $\lambda > 0$ (If not, for any $\Lambda$, there exists $\lambda(\Lambda)$ such that $d_\infty^H(\mathcal{Z}^\Lambda(\lambda(\Lambda)), \tilde{\mathcal{Z}}) <$

$\epsilon_0$, and taking $\lambda' = \min_{\Lambda \in \Xi} \lambda(\Lambda)$ we get $d_\infty^H(\mathcal{Z}(\lambda'), \tilde{\mathcal{Z}}) = \max_{\Lambda \in \Xi} d_\infty^H(\mathcal{Z}^\Lambda(\lambda'), \tilde{\mathcal{Z}}) \leq \max_{\Lambda \in \Xi} d_\infty^H(\mathcal{Z}^\Lambda(\lambda(\Lambda)), \tilde{\mathcal{Z}}) < \epsilon_0$, contradicting the assumption).

Denote for this particular $\Psi$, $U^\Psi(\lambda) := \{z \in \mathcal{Z}^\Psi(\lambda) \mid d_\infty(z, \tilde{\mathcal{Z}}) \geq \epsilon_0\} \neq \emptyset$. Notice that $U^\Psi(\lambda)$ is compact as it is both closed and bounded, and decreasing with respect to $\lambda$. Therefore, let $\lambda_i \downarrow 0$ be any decreasing series to 0, and from Cantor's intersection theorem, we have

$$\cap_{i=0}^\infty U^\Psi(\lambda_i) \neq \emptyset.$$

Note that for any $z \in \cap_{i=0}^\infty U^\Psi(\lambda_i)$, it is clear that $(x - G(z))_\Psi = 0$, i.e., $\|x - G(z)\|_0 \leq \tilde{n}$ thus $z \in \tilde{\mathcal{Z}}$, a contradiction to $d_\infty(z, \tilde{\mathcal{Z}}) \geq \epsilon_0$.

Finally, as we have for any $\hat{z}(\lambda)$ optimal to Equation 3, as $\hat{z}(\lambda) \in \mathcal{Z}(\lambda)$ we have $0 \leq d_\infty(\hat{z}(\lambda), \tilde{\mathcal{Z}}) \leq d_\infty^H(\mathcal{Z}(\lambda), \tilde{\mathcal{Z}}) \downarrow 0$ as $\lambda \downarrow 0$, or $d_\infty(\hat{z}(\lambda), \tilde{\mathcal{Z}}) \downarrow 0$ as $\lambda \downarrow 0$, which completes the proof.

### B.2 PROOF TO THEOREM 2

$\tilde{\mathcal{M}} \subseteq \{M \in \{0, 1\}^{m \times n} \mid \|M\|_0 \leq \tilde{n}\}$ comes straightforward from the definition of $\tilde{\mathcal{Z}}$. We now decompose $\tilde{\mathcal{Z}}$ in the same fashion as $\mathcal{Z}(\lambda)$. For any $\Lambda \in \Xi$ let $\tilde{\mathcal{Z}}^\Lambda := \{z \in \mathbb{S}^d \mid (x - G(z))_{ij} = 0, \forall (i, j) \notin \Lambda\}$, and we have $\cup_{\Lambda \in \Xi} \tilde{\mathcal{Z}}^\Lambda = \{z \in \mathbb{S} \mid \|x - G(z)\|_0 \leq \tilde{n}\} = \tilde{\mathcal{Z}}$, the last quality from the minimality of $\tilde{n}$. For the same reason, $\tilde{\mathcal{Z}}^\Lambda$ is empty unless $|\Lambda| = \tilde{n}$. Note that $\tilde{\mathcal{Z}}^\Lambda$ is closed and thus compact following the continuity of $G$, and thus the compactness of $\tilde{\mathcal{Z}}$.

For any non-empty $\tilde{\mathcal{Z}}^\Lambda$ and $(i, j) \in \Lambda$ we have $\inf_{z \in \tilde{\mathcal{Z}}^\Lambda} |(x - G(z))_{ij}| > 0$. If not, following the continuity of $G$, there exists $z \in \tilde{\mathcal{Z}}^\Lambda$ such that $(x - G(z))_{ij} = 0$, so $\|x - G(z)\|_0 \leq \tilde{n} - 1$, a contradiction to the minimality of $\tilde{n}$. (Note that in the case of $\tilde{\mathcal{Z}}^\Lambda = \emptyset$, $\inf_{z \in \tilde{\mathcal{Z}}^\Lambda} |(x - G(z))_{ij}| = +\infty$). Denote $s := \min_{\Lambda \in \Xi} \min_{(i,j) \in \Lambda} \inf_{z \in \tilde{\mathcal{Z}}^\Lambda} |x - G(z)|_{ij} > 0$, which is independent from $\lambda$.

Given the continuity of $G$, for any $\epsilon > 0$, there exists $r > 0$ such that for any $z, z' \in \mathbb{S}^d$ with $\|z - z'\|_\infty < r$, we have $\|G(z) - G(z')\|_\infty < \epsilon$. Specifically, we consider $s/2$ as $\epsilon$ and have the corresponding $r_{s/2}$, and we select $\tilde{\lambda} > 0$ satisfying

$$d_\infty^H(\mathcal{Z}(\tilde{\lambda}), \tilde{\mathcal{Z}}) \leq \frac{r_{s/2}}{2} \quad \text{and} \quad \tilde{\lambda} \leq \frac{s^2}{3(\tilde{n} + 1)} < \frac{s^2}{2}.$$

Notice such a $\tilde{\lambda}$ exists, since $d_\infty^H(\mathcal{Z}(\tilde{\lambda}), \tilde{\mathcal{Z}}) \downarrow 0$ as $\lambda \downarrow 0$.

For any $\lambda \leq \tilde{\lambda}$, consider any optimal solution of Equation 3 as $(\hat{z}(\lambda), \hat{M}(\lambda))$ and we have there exists $\tilde{z} \in \tilde{\mathcal{Z}}$ such that $\|\hat{z}(\lambda) - \hat{z}\|_\infty = d_\infty(\hat{z}(\lambda), \tilde{\mathcal{Z}})$ from the compactness of $\tilde{\mathcal{Z}}$. Note that

$$\|\hat{z}(\lambda) - \hat{z}\|_\infty = d_\infty(\hat{z}(\lambda), \tilde{\mathcal{Z}}) \leq d_\infty^H(\mathcal{Z}(\lambda), \tilde{\mathcal{Z}}) \leq d_\infty^H(\mathcal{Z}(\tilde{\lambda}), \tilde{\mathcal{Z}}) \leq r_{s/2}/2,$$

and thus $\|G(\hat{z}) - G(\tilde{z})\|_\infty < s/2$. As $\tilde{z} \in \tilde{\mathcal{Z}}$, there exists $\Lambda \in \Xi$ such that $\tilde{z} \in \tilde{\mathcal{Z}}^\Lambda$, noting that $|\Lambda| = \tilde{n}$. For any $(i, j) \in \Lambda$, it is clear that $|(x - G(\hat{z}(\lambda)))_{ij}| \geq |(x - G(\tilde{z}))_{ij}| - |(G(\hat{z}(\lambda) - G(\tilde{z}))_{ij}| \geq s - s/2 = s/2$. Therefore, we have $\lambda < 2(x - G(\hat{z}(\lambda)))_{ij}^2$.

Therefore, for any $(i, j) \in \Lambda$,

$$1 \geq \hat{M}(\lambda)_{ij} = 1 - \frac{\lambda}{2(x - G(\hat{z}(\lambda)))_{ij}^2} \geq 1 - \frac{2\lambda}{s^2} \uparrow 1, \quad \text{as } \lambda \downarrow 0.$$

Note that $\hat{M}(\lambda)_{ij} > 0$. We also have

$$f_{ij}(\hat{z}(\lambda), \hat{M}(\lambda); \lambda) = \lambda - \frac{\lambda^2}{4(x - G(\hat{z}(\lambda)))_{ij}^2} \geq \lambda - \frac{\lambda^2}{s^2} \geq \lambda - \frac{\lambda}{3(\tilde{n} + 1)}$$

where the last inequality from $\lambda \leq \tilde{\lambda} \leq s^2/3(\tilde{n} + 1)$.

Next, we prove that for any $(i, j) \notin \Lambda$, we have $\hat{M}(\lambda)_{ij} = 0$. Assuming there exists $(i', j') \notin \Lambda$ such that $\hat{M}(\lambda)_{i'j'} \neq 0$, we have $2(x - G(\hat{z}(\lambda)))_{i'j'}^2 \geq \lambda$ and thus

$$f_{i'j'}(\hat{z}(\lambda), \hat{M}(\lambda); \lambda) = \lambda - \frac{\lambda^2}{4(x - G(\hat{z}(\lambda)))_{i'j'}^2} \geq \frac{\lambda}{2}.$$

Therefore,

$$f(\hat{z}(\lambda), \hat{M}(\lambda); \lambda) = \sum_{(i,j)\in\Lambda} f_{ij}(\hat{z}(\lambda), \hat{M}(\lambda); \lambda) + \sum_{(i,j)\notin\Lambda} f_{ij}(\hat{z}(\lambda), \hat{M}(\lambda); \lambda)$$

$$\geq \tilde{n}(\lambda - \frac{\lambda}{3(\tilde{n}+1)}) + \frac{\lambda}{2} > \tilde{n}\lambda,$$

which is a contradiction to the optimality of $(\hat{z}(\lambda), \hat{M}(\lambda))$ since $f(\tilde{z}, \tilde{M}; \lambda) = \lambda\tilde{n}$. Therefore, for any $(i,j) \notin \Lambda$, we have $\hat{M}(\lambda)_{ij} = 0$.

In conclusion, let $\tilde{M} = I_{x-G(\tilde{z})} \in \tilde{\mathcal{M}}$ and we have $d_\infty(\hat{M}(\lambda), \tilde{\mathcal{M}}) \leq \|\hat{M}(\lambda), \tilde{M}\|_\infty \leq 2\lambda/s^2 \downarrow 0$ as $\lambda \downarrow 0$. It is also clear that $\tilde{M} = I_{\hat{M}(\lambda)}$ as long as $\lambda < \tilde{\lambda}$, which completes the proof.

## C  SIMULATION STUDY

In this section, we verify the robustness of the RGI method under gross corruptions using simulation.

**Data Generation.** A Progressive GAN (Karras et al., 2017) network is trained on the training set of 200599 aligned face images of size $128 \times 128$ from CelebFaces Attributes dataset (CelebA (Liu et al., 2015)) and the pre-trained generator $G(\cdot)$ is extracted. Then we generate a test image $x$ with central block corruptions ($\approx 25\%$ pixels) by: (i) Sample $z \in \mathbb{R}^{500}$ from the multivariate standard normal distribution, i.e., $z ~ N(0, I)$; (ii) Generate $x$ by $x_{ij} = \begin{cases} e_{ij}, & \text{if } i,j \in \{33, \dots, 96\} \\ G(z)_{ij}, & \text{otherwise} \end{cases}$, where $e_{ij} \sim N(e, 1)$ and $e$ is the mean corruption level. The pixel values of images generated by $G(\cdot)$ are approximately between $[-1, 1]$. To verify the robustness of the RGI method, we vary the mean corruption level $e$ in the range of $\{-1, -0.5, 0, 0.5, 1\}$. The process is repeated to generate 100 input corrupted images for each mean corruption level.

**Solution Procedure** For each mean corruption level, we use three methods to restore $G(z)$: (i) $l_2$: Solving Equation 1 with $l_2$ reconstruction loss $L_{rec}(\cdot) = \|\cdot\|_2^2$; (ii) $l_1$: Solving Equation 1 with $l_1$ reconstruction loss $L_{rec}(\cdot) = \|\cdot\|_1$; (iii) RGI: Solving Equation 3 with $l_2$ reconstruction loss $L_{rec}(\cdot) = \|\cdot\|_2^2$. All methods are solved by ADAM (Kingma & Ba, 2014) for 1000 iterations. The root mean squared image restoration error (RMSE) of 100 input images is recorded, i.e.,

$$\text{RMSE} = \sqrt{\frac{1}{100}\sum_{i=1}^{100} \frac{1}{mn}\|G(z_i) - G(\hat{z}_i)\|_2^2},$$

where $G(z)$ and $G(\hat{z})$ are the true and restored backgrounds, respectively.

**Results:** The RMSE of image restoration results under different corruption levels from methods (i)-(iii) is shown in Figure 5. The RGI method demonstrated superior robustness with an RMSE close to zero with respect to all five different corruption levels. $l_2$ and $l_1$ reconstruction losses perform significantly worse under large corruption magnitude, which is expected since they seek an image on the learned manifold that is close to the input image (even though $l_1$ reconstruction loss adds to the robustness a little bit), which can lead to significant deviation of the image restoration.

## D  DETAILED DISCUSSION ON SEMANTIC IMAGE INPAINTING

### D.1  IMPLEMENTATION DETAILS

A Progressive GAN (Karras et al., 2017) network is used as the backbone network. Notice that the discriminator network is usually used to regularize the generated image, such that the generated image looks real. Different methods incorporate the discriminator differently. It can either be incorporated as a separate penalty term in the objective function (Yeh et al., 2017), incorporated as a modified reconstruction loss term (Pan et al., 2021) or ignored in the loss function (Gu et al., 2020). For a fair comparison, we use a weighted combination of an $l_2$ norm (with weight 1) and a discriminator penalty term (with weight 0.1) as the reconstruction loss for comparison methods. In all

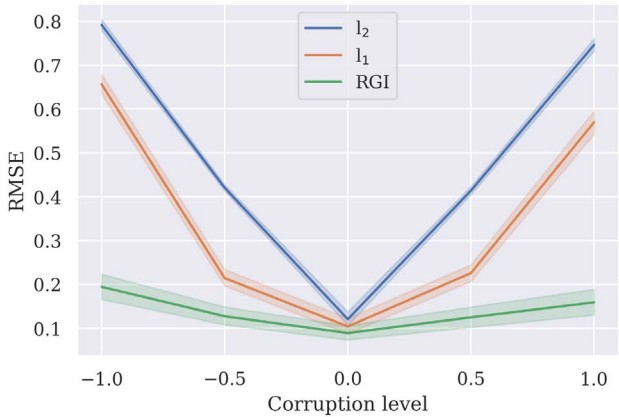

Figure 5: RMSE of image restoration by RGI and GAN-inversion with $l_1$, $l_2$ loss.

experiments, the optimization problems Equation 1 (GAN-inversion), Equation 3 (RGI) and Equation 4 (R-RGI) are solved by ADAM (Kingma & Ba, 2014) for 2000 iterations, with a learning rate of $0.1$ for both $z$ and $M$. For Equation 4 (R-RGI), we use the last 500 iterations for mask-free fine-tuning with the learning rate of $1e^{-5}$ for $\theta$. The tuning parameter $\lambda$ is selected by cross-validation. Notice that using SSIM and PSNR gives different corss-validation results and here we report both:

(i) CelebA: block missing RGI and R-RGI: $\lambda_{SSIM} = \lambda_{PSNR} = 0.07$; random missing RGI: $\lambda_{SSIM} = 0.2$, $\lambda_{PSNR} = 0.5$, R-RGI: $\lambda_{SSIM} = 0.25$, $\lambda_{PSNR} = 0.6$.

(ii) Stanford cars: block missing RGI $\lambda_{SSIM} = 0.9$, $\lambda_{PSNR} = 1.0$, R-RGI: $\lambda_{SSIM} = \lambda_{PSNR} = 0.9$; random missing RGI: $\lambda_{SSIM} = \lambda_{PSNR} = 1.0$, R-RGI: $\lambda_{SSIM} = \lambda_{PSNR} = 0.8$.

(iii) LSUN bedroom: block missing RGI: $\lambda_{SSIM} = \lambda_{PSNR} = 0.8$, R-RGI: $\lambda_{SSIM} = \lambda_{PSNR} = 0.7$; random missing RGI: $\lambda_{SSIM} = 0.8$, $\lambda_{PSNR} = 1.0$, R-RGI: $\lambda_{SSIM} = 0.6$, $\lambda_{PSNR} = 0.9$.

## D.2 DATASETS DETAILS

**CelebA (Liu et al., 2015)** contains a training set of 200,599 aligned face images. We resize them to the size of $128 \times 128$. We use the remaining 2000 images as the test set. Missing regions are generated as follows: (i) central block missing of size $32 \times 32$ and (ii) random missing ($\approx 50\%$ pixels). We fill in the missing entry with pixels from $N(-1, 1)$. We randomly select 100 test images to evaluate algorithm performance.

**Stanford cars (Krause et al., 2013)** contains 16,185 images of 196 classes of cars and is split into 8,144 training images and 8,041 testing images. We crop the image based on the provided bounding boxes and resize them to the size of $128 \times 128$. Missing regions are generated as follows: (i) central block missing of size $16 \times 16$ and (ii) random missing ($\approx 25\%$ pixels). We fill in the missing entry with pixels from $N(-1, 1)$. The training and test set partitions provided by the dataset are used. We randomly select 100 test images to evaluate algorithm performance.

**LSUN bedroom (Yu et al., 2015)** contains 3,033,042 images for training and 300 images for validation. We resize the images to the size of $128 \times 128$. Missing regions are generated as follows: (i) central block missing of size $16 \times 16$ and (ii) random missing ($\approx 25\%$ pixels). We fill in the missing entry with pixels from $N(-1, 1)$. We randomly select 100 images from the validation set to evaluate algorithm performance.

Next, we will show the qualitative image restoration results on these datasets. Notice that we avoid showing the CelebA result due to copyright/privacy concerns.

## D.3 QUALITATIVE IMAGE RESTORATION RESULTS

Figure 6 shows the qualitative image restoration results on Stanford cars (Krause et al., 2013) dataset. From columns 2-4, we can observe that the RGI method has a comparable performance as (Yeh

et al., 2017) w/ mask, which improves the image restoration performance of (Yeh et al., 2017) w/o mask. However, the performance of Yeh et al. (2017) even with mask information is not satisfactory (for example, the second row of Figure 6), this is mainly due to the GAN approximation gap (Pan et al., 2021). In this case, further generator fine-tuning will significantly improve the faithfulness of restored images, which can be observed from columns 5-6.

Figure 7shows the qualitative image restoration results on the LSUN bedroom (Yu et al., 2015) dataset. A similar conclusion can also be drawn.

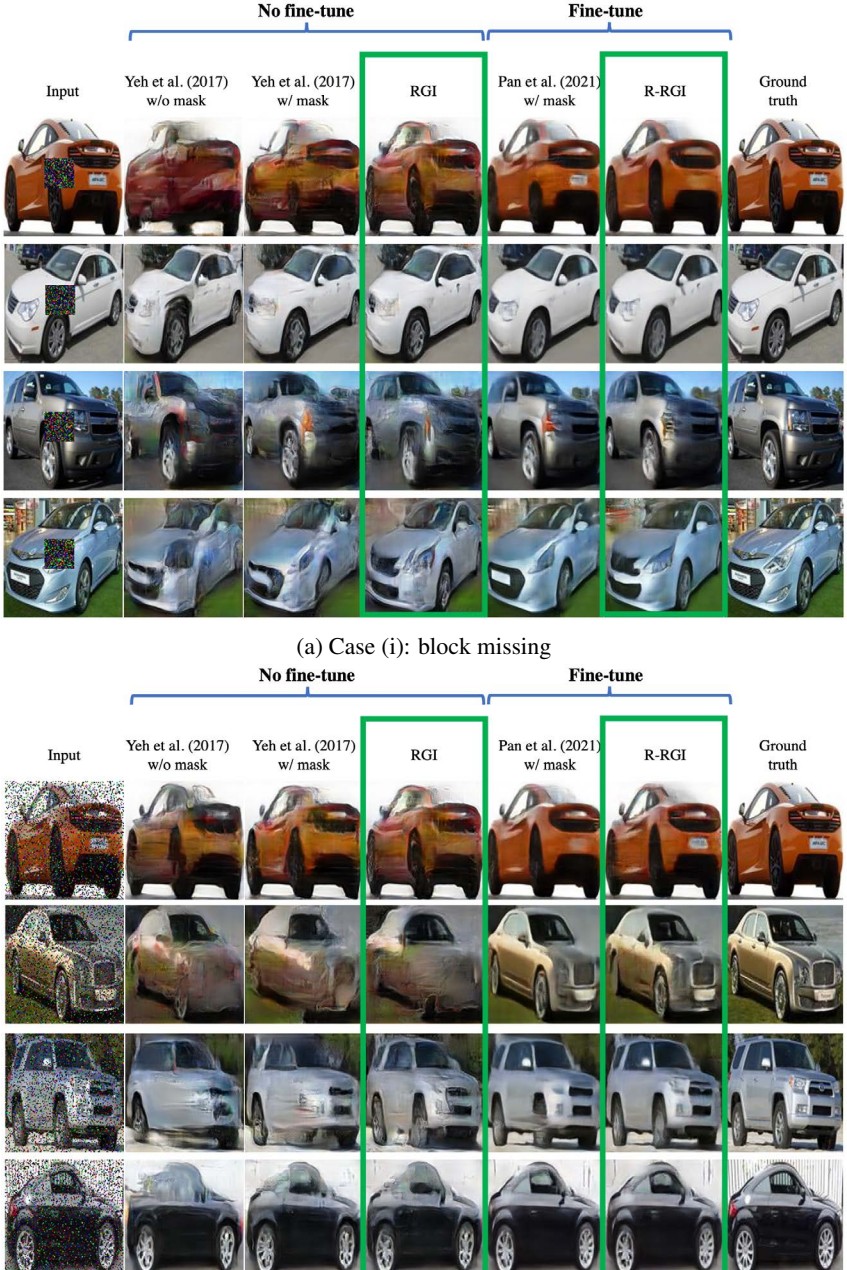

(a) Case (i): block missing

(b) Case (ii): random missing

Figure 6: Restored images by Yeh et al. (2017) (w/ and w/o mask), Pan et al. (2021) (w/ mask), RGI and R-RGI on Stanford cars (Krause et al., 2013) dataset.

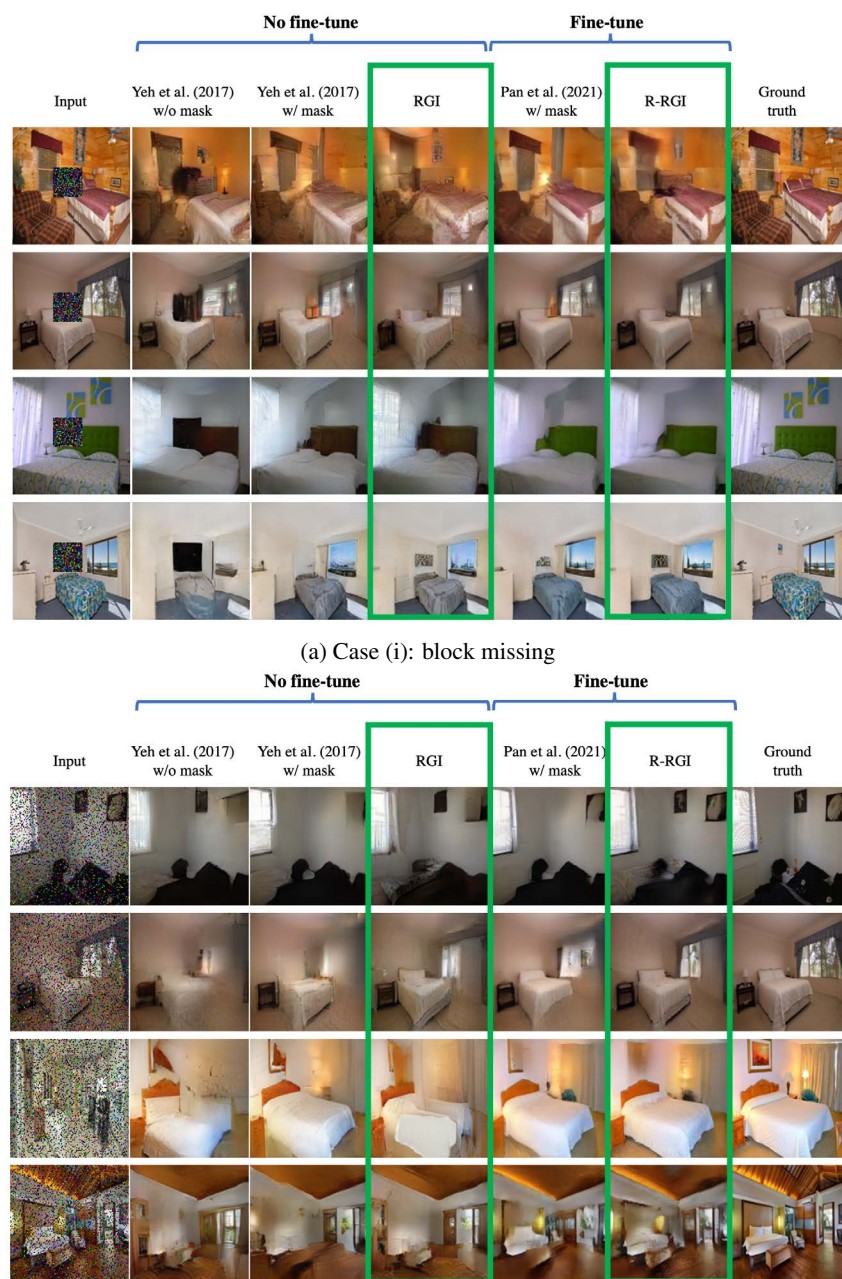

(a) Case (i): block missing

(b) Case (ii): random missing

Figure 7: Restored images by Yeh et al. (2017) (w/ and w/o mask), Pan et al. (2021) (w/ mask), RGI and R-RGI on LSUN bedroom (Yu et al., 2015) dataset.

# E    DETAILED RESULTS ON THE MVTEC AD DATASET

**Annotation issues of the MVTec AD dataset**. Figure 8 shows example images from the MVTec dataset as well as the corresponding annotations. It is clear that the annotation covers a larger area than the exact defect contour, which will favor localization level methods such as PatchCore (Roth et al., 2022). However, this level of annotation is neither sufficient to fulfill the fine-grained surface quality inspection goal, such as providing precise defect specifications (i.e. diameter, length, area) for product surface quality screening, nor serve as an effective dataset for training/evaluating pixel-wise anomaly detection algorithms.

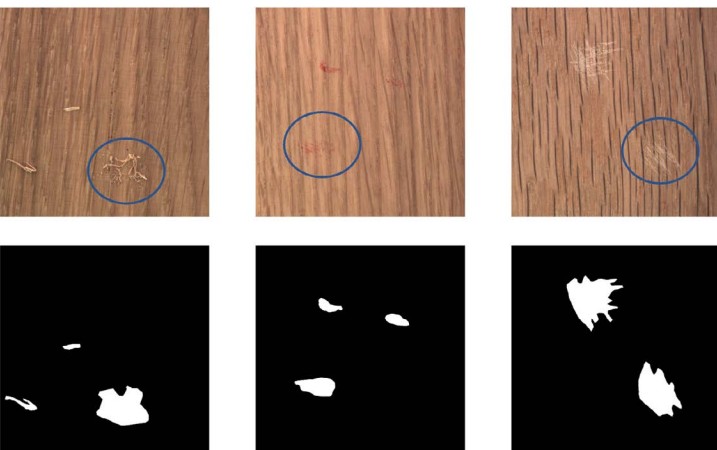

Figure 8: Example defective products and their annotations from the MVTec AD dataset: The blue circle indicates the defects. we can see that the annotation tends to cover extra regions of the real anomaly contour.

**Qualitative assessment on MVTec AD.** Figure 9 shows the qualitative results of the wood product. We can observe that both the performance of RGI and AnoGAN are poor, where the restored images are far from the true background, which leads to a noisy anomalous segmentation mask. The main reason is the small size of the training set, where the learned generator tends to overfit (memorize) the training set (Karras et al., 2020; Webster et al., 2019) rather than generalize to images in the test set. This will lead to a huge gap between the learned training image manifold and the testing image manifold. By generator fine-tuning, the R-RGI can mitigate this gap to improve both the background reconstruction and anomalous region identification performance.

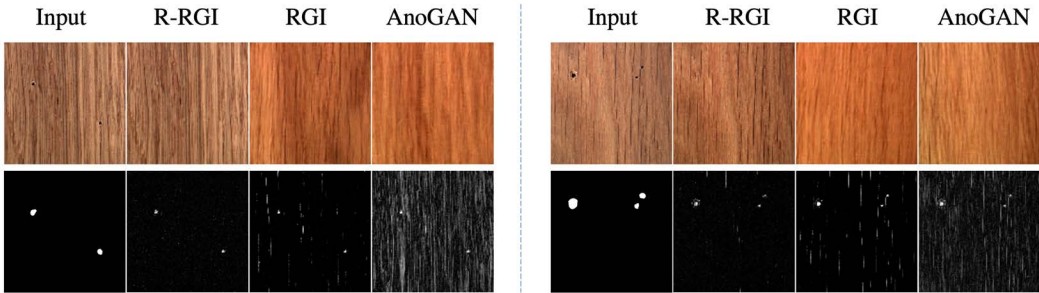

Figure 9: Comparison between the restored backgrounds and detected anomalies on wood product: the first row indicates the input image and reconstructed background by various methods; the second row indicates the defective region segmentation mask annotation from the MVTec AD dataset and the detected anomalous region segmentation mask by various methods.

However, the success of the RGI/R-RGI method is built upon the assumption of a large training dataset such that the generator can learn a reasonable manifold which can generalize to unseen test samples. Mask-free fine-tuning can then mitigate the GAN approximation gap to further improve its performance. When the training set size is too small, where the generator tends to overfit (memorize) the training set, merely relying on the mask-free fine-tuning can lead to an unstable result.

## F    SYNTHETIC DEFECT GENERATION ON BTAD PRODUCT03

The detailed defect generation process is discussed in this section. Product03 has 1000 defective free images, from which we randomly select 100 images for defect generation. To improve the

faithfulness of generated defective images, we collect the binary defective region masks (equals to 1 for defective pixels and 0 otherwise) from the annotations of the MVTec AD (Bergmann et al., 2019) dataset and organize them into 4 categories, including crack, irregular, scratch, and mixed large (defective region area larger than 400 pixels) type of defective region masks. Then, we generate the synthetic defective image $x_i^{sys,j}$ by:

$$x_i^{sys,j} = (1 - M_i^j) \odot x_i + M_i^j \odot C_i^j, i \in [1, ...100], j \in \{\text{crack, irregular, scratch, mixed large}\},$$

where $x_i$ is the $i$th input defect-free image, $M_i^j$ is the $i$th randomly selected mask from the $j$th category. $C_i^j \in \mathbb{R}^{m \times n \times 3}$ is an image with constant channel values to fill in the defective region. To avoid trivial anomaly detection, we set $C_i^j$ as the average pixel value of the defective region, i.e.,

$$C[:,:,k]_i^j = \frac{\sum_{p_1 \in [m], p_2 \in [n]} M_i^j[p_1, p_2, k] x_i[p_1, p_2, k]}{\sum_{p_1 \in [m], p_2 \in [n]} M_i^j[p_1, p_2, k]}, k \in [3].$$

Finally we have 4 categories of synthetic defects with 100 defective images in each category. Examples of generated defective images are shown in Figure 10. We can observe that the defects are close to the background color, which makes them hard to distinguish even with human eyes. This avoids trail defect detection.

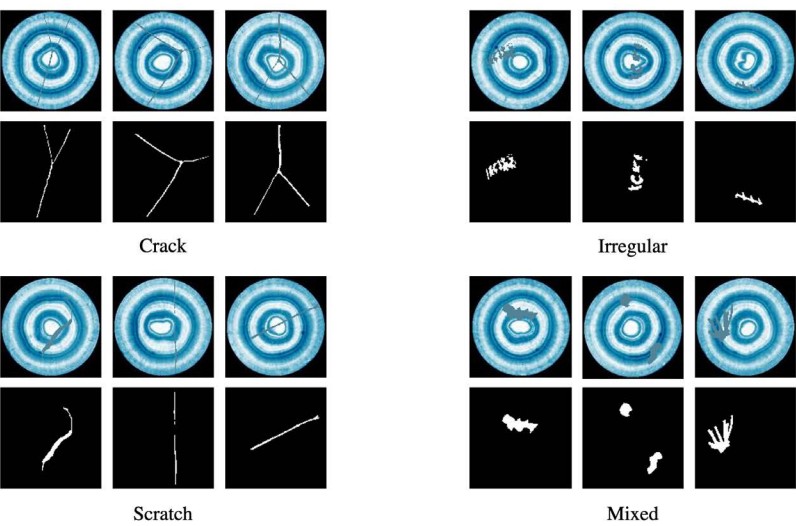

Figure 10: Generated synthetic defective images.

# G   DETAILED RESULTS ON THE BTAD DATASET

## G.1   IMPLEMENTATION DETAILS

We use a PGGAN (Karras et al., 2017) as the backbone network and a $l_2$ norm reconstruction loss term ($L_{rec}$) for RGI and R-RGI methods. The tuning parameter $\lambda$ is selected via cross validation by using Dice coefficient as the metric. For RGI, the following values are selected: $\lambda_{crack} = \lambda_{irregular} = \lambda_{scratch} = \lambda_{mixed\ large} = 0.4$. For R-RGI, the following values are selected: $\lambda_{crack} = 0.12$, $\lambda_{irregular} = 0.1$, $\lambda_{scratch} = 0.14$, $\lambda_{mixed\ large} = 0.06$. All optimization problems are solved by ADAM (Kingma & Ba, 2014) for 2000 iterations, with a learning rate of 0.1 for both $z$ and $M$. For R-RGI, we use the last 1500 iterations for mask-free fine-tuning with the learning rate of $1e^{-5}$ for $\theta$.

## G.2 QUALITATIVE RESULTS

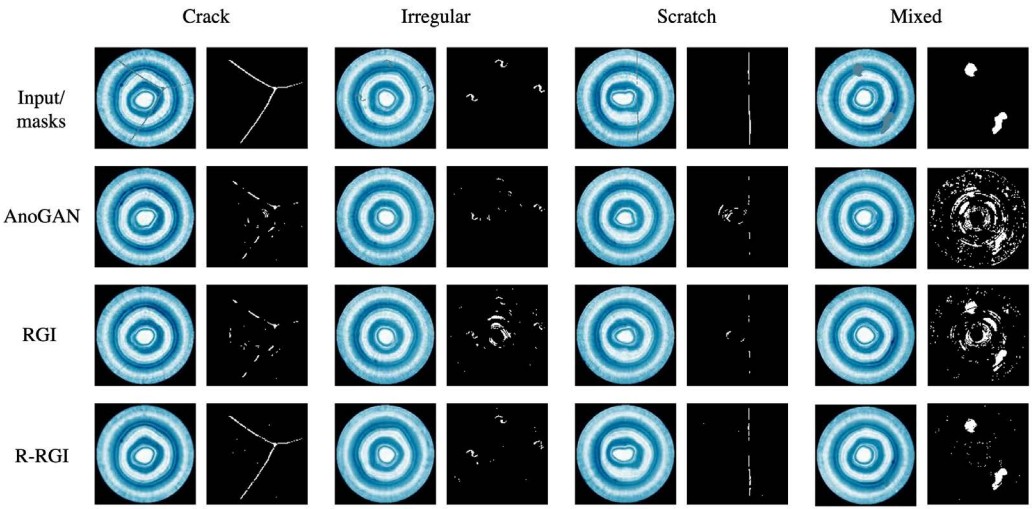

Figure 11: Comparison between the restored backgrounds and detected anomalies (after optimal thresholding) among AnoGAN (Schlegl et al., 2017), RGI and R-RGI: The first row indicates the input images and true anomalous region segmentation masks. The following rows are restored backgrounds and detected anomalous region masks by different methods.

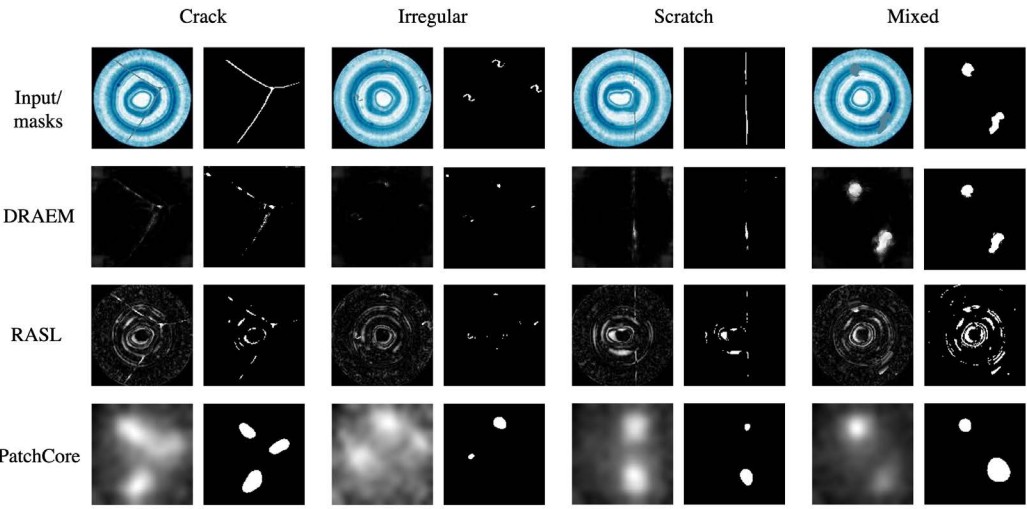

Figure 12: Raw anomaly scores and detected anomalies (after optimal thresholding) among DRAEM (Zavrtanik et al., 2021), RASL (Peng et al., 2012) and PatchCore (Roth et al., 2022): The first row indicates the input images and true anomalous region segmentation masks. The following rows are raw anomaly scores and detected anomalous region masks (after optimal thresholding) by different methods.

## H  APPLICATION ASPECT OF THE PROPOSED METHOD

In this section, we will discuss the application aspect of the proposed method.

### H.1  COMPUTATIONAL ASPECT

As mentioned in Section 2, RGI/R-RGI is a first attempt to address the current robustness and approximation gap issue of GAN-inversion. Therefore, we choose to study the optimization-based GAN-inversion methods, because of its popularity and significant advantage of superior image restoration quality (Zhu et al., 2020; Richardson et al., 2021; Creswell & Bharath, 2018). However, the proposed RGI/R-RGI method faces a similar computational challenge as optimization-based GAN-inversion methods: the inversion process of each input image requires solving an optimization problem of similar size, which can be computationally expensive. Generalizing the proposed method to learning-based as well as hybrid GAN-inversion methods for better computational efficiency is an immediate extension, and we leave it for future work.

### H.2  REDUCING THE TRAINING SAMPLE SIZE REQUIREMENT

The proposed R-RGI method can reduce the training data requirement for GAN-based anomaly detection methods (Schlegl et al., 2017; 2019). GAN-based methods (Xia et al., 2022b) are important directions in anomaly detection, which usually needs a large dataset of cleaning training images. Insufficient training samples is one of the major sources of the GAN approximation gap. Failing to address such a gap will lead to the poor performance (in addition to the robustness issue) of most of current GAN-inversion based anomaly detection methods (Xia et al., 2022b; Zenati et al., 2018; Schlegl et al., 2019; Baur et al., 2018; Kimura et al., 2020; Akcay et al., 2018).

As can be seen from the experiment result in unsupervised anomaly detection (Table 2), the performance improvement of RGI over the AnoGAN is still limited, which indicates a large GAN approximation gap, i.e., there is still a large mismatch between the closest image in the learned image manifold and the ground truth background of the input test image, beyond the robustness issue. By addressing such a gap in R-RGI, we observed a significant performance improvement in all 4 different defect scenarios, and the R-RGI outperforms the SOTA method (DRAEM) on this task.

## I  MASK-FREE IMAGE INPAINTING OF IRREGULAR MISSING REGIONS

In this section, we include the mask-free semantic image inpainting result when the missing region is of irregular shapes, as missing case (iii). The irregular missing masks are from Irregular Mask Dataset (Liu et al., 2018). The qualitative result is shown in Figure 13 and Figure 14. The quantitative result is shown in Table 3. We can observe that the inpainting performance of irregular missing in case (iii) and block missing in case (i) are fairly close (Tables 3 and 1). This is expected since there is no mask information involved in both the training and testing stage of the proposed method, which avoids potential overfitting to any specific missing mechanism. In other words, the irregular missing, block missing or even random missing are all 'unknown gross corruptions' to the RGI/R-RGI method.

Table 3: Case (iii) irregular missing region: semantic inpainting performance of Yeh et al. (2017) (w/ and w/o mask), Pan et al. (2021) (w/ mask), RGI and R-RGI

| Datasets | Cases | metrics | methods | | | | |
|---|---|---|---|---|---|---|---|
| | | | Yeh et al. w/o mask | Yeh et al. w/ mask | RGI | Pan et al. w/ mask | R-RGI |
| CelebA | Case (iii) | PSNR ↑ | 13.72 | 21.25 | **19.86** | 22.71 | **20.42** |
| | | SSIM ↑ | 0.358 | 0.503 | **0.592** | 0.570 | **0.547** |
| Cars | Case (iii) | PSNR ↑ | 17.07 | 17.57 | **17.04** | 21.34 | **20.20** |
| | | SSIM ↑ | 0.359 | 0.361 | **0.357** | 0.633 | **0.613** |
| LSUN bedroom | Case (iii) | PSNR ↑ | 17.52 | 19.00 | **18.20** | 21.30 | **19.77** |
| | | SSIM ↑ | 0.393 | 0.424 | **0.412** | 0.599 | **0.575** |

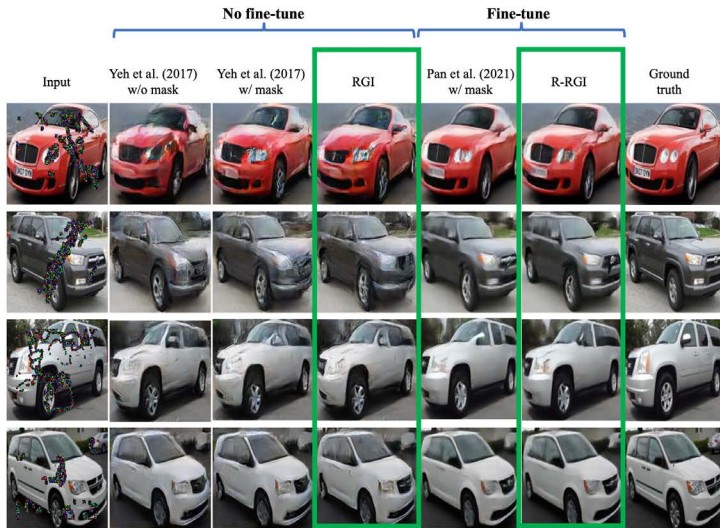

Figure 13: Case (iii) irregular missing region: restored images by Yeh et al. (2017) (w/ and w/o mask), Pan et al. (2021) (w/ mask), RGI and R-RGI on Stanford cars (Krause et al., 2013) dataset.

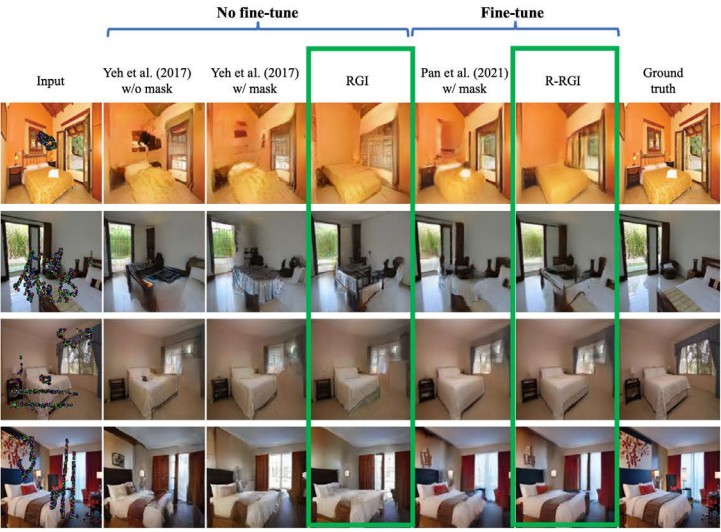

Figure 14: Case (iii) irregular missing region: restored images by Yeh et al. (2017) (w/ and w/o mask), Pan et al. (2021) (w/ mask), RGI and R-RGI on LSUN bedroom (Yu et al., 2015) dataset.

## J  SENSITIVITY ANALYSIS OF $\lambda$

The tuning parameter $\lambda$ in RGI/R-RGI (equation 3 and equation 4) is important in both mask-free semantic inpainting and unsupervised pixel-wise anomaly detection applications. In this section, we provide the empirical sensitivity analysis using examples from both applications.

**Sensitivity analysis of $\lambda$ in mask-free semantic image inpainting.** Figure 15 shows the SSIM value change with respect to $\lambda$ in mask-free semantic inpainting of block missing experiment on CelebA (Liu et al., 2015) dataset. In this experiment, we vary the tuning parameter $\lambda$ in the

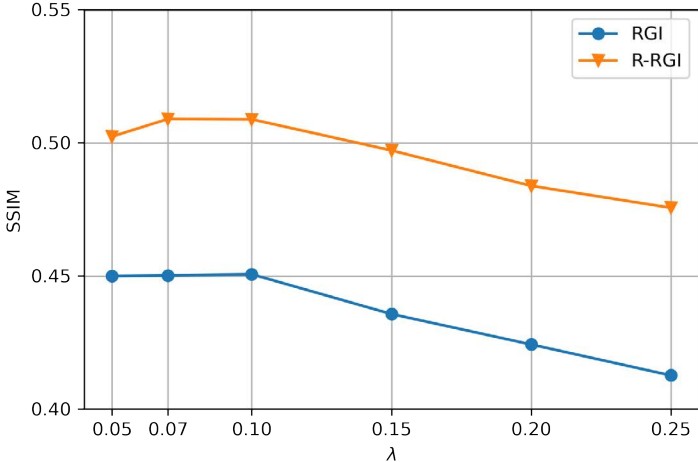

Figure 15: SSIM value change with respect to $\lambda$ in mask-free semantic inpainting of block missing experiment on CelebA (Liu et al., 2015) dataset.

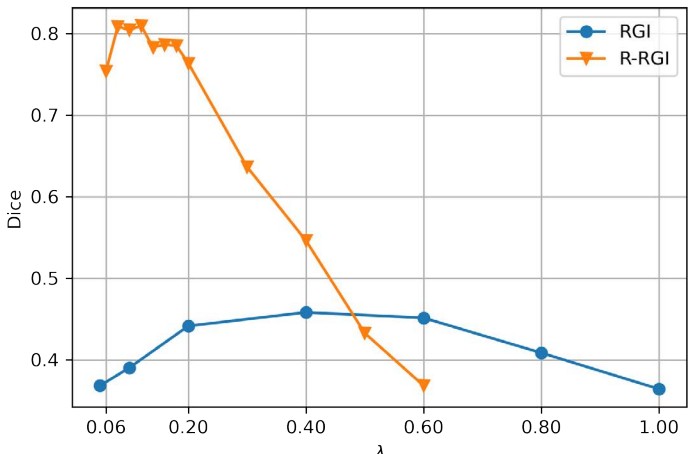

Figure 16: Dice value change with respect to $\lambda$ in unsupervised pixel-wise anomaly detection of crack defect experiment.

range of $[0.05, 0.25]$. A plateau region can be observed for both the RGI ($[0.05, 0.10]$) and R-RGI ($[0.05, 0.15]$) method, where similar inpainting performance as the optimal $\lambda^*$ can be obtained.

**Sensitivity analysis of $\lambda$ in unsupervised pixel-wise anomaly detection.** Figure 16 shows the Dice value change with respect to $\lambda$ in in unsupervised pixel-wise anomaly detection of crack defect experiment on the synthetic dataset. For RGI, we vary the tuning parameter $\lambda$ in the range of $[0.05, 1.00]$. A large plateau region ($[0.20, 0.60]$) can be observed, where similar semantic inpainting performance as the optimal $\lambda^*$ can be obtained. For R-RGI, we vary the tuning parameter $\lambda$ in the range of $[0.06, 0.6]$. Compared to RGI, R-RGI has an additional generator fine-tuing step when optimizing the mask $M$, which makes R-RGI more sensitive to $\lambda$. However, we can still observe a plateau region ($[0.06, 0.2]$), where acceptable semantic inpainting performance, compared to the optimal $\lambda^*$, can be obtained. The existence of such a plateau region in both applications makes it easier for the tuning of $\lambda$.

