# OpenReview forum: "RGI: robust GAN-inversion for mask-free image inpainting and unsupervised pixel-wise anomaly detection"
_ICLR.cc/2023/Conference — ICLR 2023 poster_

### Official Review · Reviewer_XsAp · 2022-10-22

**Confidence:** 3
**Clarity, Quality, Novelty And Reproducibility:** 1. We all agree that image inpainting…
**Correctness:** 3
**Technical Novelty And Significance:** 2
**Empirical Novelty And Significance:** 2
**Recommendation:** 5

**Details Of Ethics Concerns:**

I cannot find ethics concerns of this work. However, if the method are used for face images, it may lead to some social problems.

**Strength And Weaknesses:**

The strength and weakness of this problem are all very obvious.
1. By using the GAN-Inversion, the author achieves better results on non-mask image inpainting scenarios, which help the classical image inpainting problems overcome a challenging obstacle.
2. I know that for image inpainting problems, the guys always overfit their module some specified scenarios, and show very promising results on some test data with similar properties. Yeh etal is just like this. I am not sure whether this method is still in this way. If they still solve this problem in this manner and show poor generalization ability to other scenarios, I donot think the GAN-Inversion is a big plus.
3. To relive my concern, please provide more details in the rebuttal about the details of training data, for example, if you want to get the result on CELEBA dataset, do you need to use similar data to train your network for the generative image prior?
4. Is it ok to show more results beyond current established dataset?
5. Is it ok to provide more details about Figure 2? From the figure, this work still relies on some kinds of mask to achieve the results, though this mask is not explicitly obtained. However, considering the examples in this figure, where the mask is with significant differences to its surrounding regions, it may not be very difficult to estimate such a mask.
6. Image inpainting has already been a rat race. Many methods have been proposed. Why do you only compare with so limited methods?

**Summary Of The Paper:**

This paper introduces a GAN-Inversion method to solve the image restoration problems. From the results shown in this paper, this author emphazie a lot on the image inpainting problem, and also show the potential in solving the anomaly detection problems.

**Summary Of The Review:**

Well. I am also very busy recently, thus I may not have enough time to go through every detail of this paper. However, based on my experiences in the past several years, this paper still needs some time to be better refined.

---

> ### Comment · Reviewer_sD9p · 2022-11-05
> **Respectfully disagree with your statement.**
>
> As a reviewer, I respectfully disagree with some of your statements.
>
> According to your comments:
> 1. We all agree that image inpainting is not a new problem.
> 2. We all agree that gan-inversion technique is not a new method.
> 3. Thus using 2 to solve 1 is actually an increanmental improvement.
>
> Every work that attempts to apply an existing method to a known task is ``incremental''. This covers most of the published papers. I don't think this is a reasonable way to criticize the paper.
>
> As your comments:
> Well. I am also very busy recently, thus I may not have enough time to go through every detail of this paper.
>
> This will only undermine the legitimacy of your opinion. As a reviewer, you should at least fully examine the body of the article. I understand that reviewer efforts are often a form of voluntary labor, but it is a necessary job and service that sustains our community. I think these statements of yours are inappropriate.

---

> > ### Comment · Reviewer_XsAp · 2022-11-05
> > **Thanks for your feedback to my comments**
> >
> > I have examined the body of the article. I only say that I donot have enought time go through every detail of the article. This does not mean I do not examine the body of the article. All comments I made are all based on my feelings after I read the body of the article. The article review took me two hours. However, since I do not have enough time to carefully consider every aspect of this article, I just put that sentence in this place. I respect your view that the "incremental" comments you made were not fair to this work. However, I still think this work is incremental at this stage. I can also change my mind if the author makes more comments during the rebuttal.
> >
> > Our community depends on this job and service. In light of this, I honestly acknowledge that I didn't ckeck every detail of this article during the review for the references. Besides, this is still a paper to be reviewed. I think it is very difficult to check every detail of this paper. Even for an accepted paper, it is also not easy to go though evry detail of a paper. Thus, I think a responsible reviewer should acknowledge that they do not go through every detail of an article.
> >
> > Besides, the reviewing process at ICLR is a dynamic process. During the rebuttal process, reviewers may also deepen their understanding of an article. The comments in the first round should never be the final comments. We should only get the conclusion after we discuss with the authors.
> >
> > I have to say that I still feel the contribution of this work is somewhat "incremental". As a researcher in this field, I also do not like this word. This is also my first time to use this word.  However, since this work does not solve some image inpainting methods that should be addressed in my opinion. For example, this work still follows the classical setting when they train their models by using a large number of images with similar appearances or properties. Hope my concern can be addressed based on the authors' reponse.

---

> > > ### Comment · Reviewer_sD9p · 2022-11-05
> > > **Thanks for the clarification**
> > >
> > > Thanks for your patience and reply.
> > >
> > > In fact. I have nothing against your assessment that the article is incremental. I actually share similar feelings. I just think there's something sloppy in the way you came to that conclusion, or at least it looks sloppy. This may give the author and other bystanders an opportunity for misunderstanding. I think your comment is important, but the way you comment is not easily accepted.
> > >
> > > Thanks for the clarification. I agree that "responsible reviewers should admit that they do not scrutinize every detail of the article". Indeed when you really don't have a limited amount of time to be sure that your review is completely correct. I apologise to you for that.

---

> ### Author Response · Authors · 2022-11-17
> **Response to Reviewer XsAp, Part I.**
>
> # A concise outline of our contribution
> We deeply appreciate that the reviewer spends precious time reviewing our paper even when the reviewer is really busy. We feel sorry that our presentation failed to clearly convey our contribution to the reviewer in such a limited time period. We also feel grateful that the reviewer advocates for a dynamic reviewer process and gives us the chance to explain our work again. Therefore, please allow us to give a brief summary of the motivation and thought process when developing this method and a concise outline of our contribution.
>
> The main contribution of this work is not introducing GAN-inversion to solve image restoration problems (which are the main focus of previous works such as [1-4]). Instead, we would like to emphasize that, methodologically, this paper is a nontrivial extension of the traditional robust machine learning methods, such as the Robust Principal Component Analysis (RPCA) method, to the deep learning era by introducing a learned deep generative prior (DGP) by GAN.
>
> As mentioned in the Discussion (Section 3.3.), robust machine learning has been a popular signal processing method in the pre-deep learning era, which aims to restore a clean signal (or achieve robust parameter estimation) in the presence of corrupted input data. It has simple and intuitive problem formulation but (somehow surprisingly) successful and wide application scenarios, such as including robust signal denoising and dimensional reduction [5], dynamic foreground and static background separation, and anomaly detection [6, 7], as long as the signal satisfies the statistical assumptions of robust machine learning methods, such as low rank (RPCA), smoothness, and so on. However, we notice that in lots of applications, such assumptions can be restrictive. To address this issue, we introduce a learned deep generative prior (DGP), by GAN, to the traditional robust learning methods.
>
> By such a non-trivial extension, the proposed method not only inherits the robust nature, simple and intuitive problem formulation of robust machine learning, but also introduces a strong modeling capability. And in turn, we address the robustness and approximation gap in the traditional GAN-inversion method. Therefore, we name it Robust GAN-inversion (RGI).
>
> In this way, the proposed method can achieve good results in problems that are originally difficult by either using GAN-inversion or RPCA alone, such as mask-free semantic inpainting of complex human face images or unsupervised pixel-wise anomaly detection in complex background images from a real manufacturing line.
>
> # Point-by-point responses
> Below we provide our point-by-point responses to your detailed remarks, which we hope can address your current reservations.
>
> ## Response to weaknesses #2 and #3
> We thank the reviewer for these comments. We would like to address the reviewer’s concern on in two aspects:
> 1) **The GAN-inversion-based methods do not overfit any missing mechanism during training.** There have been a series of works on GAN-inversion for semantic inpainting [1-4]. We have a detailed discussion about current semantic image inpainting methods in Appendix A.1. As a summary, most of the current semantic image inpainting methods usually need mask information in the training stage, either the same fixed mask as the region to be inpainted [8], or trying to cover irregular missing regions by randomly sampling a rectangular mask with random location [9], or a fixed set of irregular masks [10] or generating masks following a set of rules [11], where overfitting to specific missing scenarios in their training sets can be a problem. However, compared to those methods, one main advantage of the GAN-inversion-based method is that it does not need any mask information during training. Therefore, there is no overfitting towards any specific missing mechanism (missing hole shape) and its inference is possible irrespective of how the missing content is structured [2].
> 2) **Training and inference on the same dataset (divided into training and test set) do not mean overfitting.** Yes, if we want to inpaint missing regions of test images from CELEBA dataset, we indeed need to get the deep generative prior by training our network on images from the training set of CELEBA dataset. The training and test datasets are strictly separated. Therefore, there is no overfitting.
> Generally speaking, training and inference on the same dataset (divided as training and test set) is a commonly accepted practice in GAN-Inversion literature and machine learning. (We are not referring to transfer learning related methods.) In fact, GAN-inversion is a popular technique in general image editing, even beyond semantic inpainting, to name a few [12-14]. They all require training on the same dataset and then editing images from the test set of the same dataset.

---

> > ### Author Response · Authors · 2022-11-17
> > **Response to Reviewer XsAp, Part II.**
> >
> > ## Response to weaknesses #4
> > We have covered commonly used datasets for developing and benchmarking semantic inpainting algorithms [1-3] including CelebA [15], Stanford Cars[16], and LSUN [17]. In addition, to demonstrate the performance of the proposed method, we added one case by inpainting irregular missing shapes in the revised paper.
> >
> > However, we do not quite understand what the reviewer means by “beyond currently established dataset”. We are happy to discuss this to further relieve the reviewer’s concern.
> >
> > ## Response to weaknesses #5
> > We apologize for the confusion caused by Figure 2 and we added more details according to the reviewer’s suggestion in the revised paper. The proposed algorithm does not rely on any mask information. Instead, the mask $M$ is an optimization variable to be solved, simultaneously as the latent code $z$.
> >
> > In Figure 2, the missing region is purposely filled in with pixels of large deviations from the ground truth image, which indicates a large magnitude of corruption. We use this to demonstrate that the RGI method can still restore the ground truth image even under such a challenging case.
> >
> > However, we also demonstrate the success of the proposed RGI/R-RGI method on scenarios where the corrupted region mask is difficult to estimate, such as random missing in semantic inpainting and unsupervised pixel-wise anomaly detection application. For example, in Figure 10, 11 and 12, the corruptions (defects) are even difficult to identify even by human eyes.
> >
> > ## Response to weaknesses #6
> > We agree that there have been a series of works on semantic image inpainting. However, the mask-free (in both the training and testing stage) image inpainting is a novel and challenging problem and thus, there is limited literature that covers this goal. We would like to emphasize that we have compared with available literature satisfying the mask-free semantic inpainting goal.To provide a more comprehensive result, we even compare with some directly related literature [2, 4] that do require a preconfigured segmentation mask for semantic inpainting. We demonstrated that our proposed method, even though without the preconfigured segmentation mask requirement, can achieve comparable performance as [2, 4].
> >
> > A detailed discussion on related literature can be found in the comprehensive literature review for the mask-free semantic inpainting task in Appendix A.1.
> >
> > ## Response to “Clarity, Quality, Novelty And Reproducibility”
> > ### Response to #1-3.
> > The authors do agree that “image inpainting is not a new problem” and “gan-inversion technique is not a new method”. However, the main contribution of this paper is NOT using GAN-inversion to solve image inpainting problems (which is the focus in previous works such as [1-4]).
> >
> > Instead, we developed an RGI/R-RGI method, which is a nontrivial extension of the traditional robust machine learning methods by introducing a DGP learned by GAN, and in turn, addresses the robustness and approximation gap in the traditional GAN-inversion method.
> > Then, we adopt it to solve novel and important problems that are originally difficult by either using GAN-inversion or robust machine learning methods along, such as mask-free semantic inpainting (in both training and testing stages) of complex human face images and unsupervised pixel-wise anomaly detection in the complex background, which are both important and new problems that are less addressed.
> >
> > ### Response to #4 has been covered in the Response to weaknesses #5.
> > ### Response to #5
> > We have provided the code for reproducing the results, more implementation details can be found in the code.

---

> > > ### Author Response · Authors · 2022-11-17
> > > **Response to Reviewer XsAp, Part III.**
> > >
> > > **References**
> > >
> > > [1]    	J. Gu, Y. Shen, and B. Zhou, "Image processing using multi-code gan prior," in Proceedings of the IEEE/CVF conference on computer vision and pattern recognition, 2020, pp. 3012-3021.
> > >
> > > [2]    	R. A. Yeh, C. Chen, T. Yian Lim, A. G. Schwing, M. Hasegawa-Johnson, and M. N. Do, "Semantic image inpainting with deep generative models," in Proceedings of the IEEE conference on computer vision and pattern recognition, 2017, pp. 5485-5493.
> > >
> > > [3]    	W. Wang, L. Niu, J. Zhang, X. Yang, and L. Zhang, "Dual-path Image Inpainting with Auxiliary GAN Inversion," in Proceedings of the IEEE/CVF Conference on Computer Vision and Pattern Recognition, 2022, pp. 11421-11430.
> > >
> > > [4]    	X. Pan, X. Zhan, B. Dai, D. Lin, C. C. Loy, and P. Luo, "Exploiting deep generative prior for versatile image restoration and manipulation," IEEE Transactions on Pattern Analysis and Machine Intelligence, 2021.
> > >
> > > [5]    	E. J. Candès, X. Li, Y. Ma, and J. Wright, "Robust principal component analysis?," Journal of the ACM (JACM), vol. 58, no. 3, pp. 1-37, 2011.
> > >
> > > [6]    	T. Bouwmans, A. Sobral, S. Javed, S. K. Jung, and E.-H. Zahzah, "Decomposition into low-rank plus additive matrices for background/foreground separation: A review for a comparative evaluation with a large-scale dataset," Computer Science Review, vol. 23, pp. 1-71, 2017.
> > >
> > > [7]    	T. Bouwmans and E. H. Zahzah, "Robust PCA via principal component pursuit: A review for a comparative evaluation in video surveillance," Computer Vision and Image Understanding, vol. 122, pp. 22-34, 2014.
> > >
> > > [8]    	D. Pathak, P. Krahenbuhl, J. Donahue, T. Darrell, and A. A. Efros, "Context encoders: Feature learning by inpainting," in Proceedings of the IEEE conference on computer vision and pattern recognition, 2016, pp. 2536-2544.
> > >
> > > [9]    	C. Yang, X. Lu, Z. Lin, E. Shechtman, O. Wang, and H. Li, "High-resolution image inpainting using multi-scale neural patch synthesis," in Proceedings of the IEEE conference on computer vision and pattern recognition, 2017, pp. 6721-6729.
> > >
> > > [10]  	G. Liu, F. A. Reda, K. J. Shih, T.-C. Wang, A. Tao, and B. Catanzaro, "Image inpainting for irregular holes using partial convolutions," in Proceedings of the European conference on computer vision (ECCV), 2018, pp. 85-100.
> > >
> > > [11]  	J. Yu, Z. Lin, J. Yang, X. Shen, X. Lu, and T. S. Huang, "Free-form image inpainting with gated convolution," in Proceedings of the IEEE/CVF international conference on computer vision, 2019, pp. 4471-4480.
> > >
> > > [12]  	R. Abdal, Y. Qin, and P. Wonka, "Image2stylegan++: How to edit the embedded images?," in Proceedings of the IEEE/CVF conference on computer vision and pattern recognition, 2020, pp. 8296-8305.
> > >
> > > [13]  	R. Abdal, Y. Qin, and P. Wonka, "Image2stylegan: How to embed images into the stylegan latent space?," in Proceedings of the IEEE/CVF International Conference on Computer Vision, 2019, pp. 4432-4441.
> > >
> > > [14]  	T. Karras, S. Laine, M. Aittala, J. Hellsten, J. Lehtinen, and T. Aila, "Analyzing and improving the image quality of stylegan," in Proceedings of the IEEE/CVF conference on computer vision and pattern recognition, 2020, pp. 8110-8119.
> > >
> > > [15]  	Z. Liu, P. Luo, X. Wang, and X. Tang, "Deep learning face attributes in the wild," in Proceedings of the IEEE international conference on computer vision, 2015, pp. 3730-3738.
> > >
> > > [16]  	J. Krause, M. Stark, J. Deng, and L. Fei-Fei, "3d object representations for fine-grained categorization," in Proceedings of the IEEE international conference on computer vision workshops, 2013, pp. 554-561.
> > >
> > > [17]  	F. Yu, A. Seff, Y. Zhang, S. Song, T. Funkhouser, and J. Xiao, "Lsun: Construction of a large-scale image dataset using deep learning with humans in the loop," arXiv preprint arXiv:1506.03365, 2015.

---

> > ### Comment · Reviewer_XsAp · 2022-11-23
> > **Thanks for your feedback**
> >
> > I read this paper again. From its descriptions in the experiments, the authors want to evaluate whether their method can outperform other methods in doing a mask-free training and a mask-free inference. Actually, when I read this article initially, one of my question is why it just compares with two methods. For example, this method is not considered in the experiments:
> > Yu, Jiahui, et al. "Generative image inpainting with contextual attention." Proceedings of the IEEE conference on computer vision and pattern recognition. 2018.
> > However, from the description, their settings may hinder such comparison, since some methods cannot run properly without such a mask.
> >
> > I know that this may be regarded as an advantage. However, is it impossible to find more methods to the comparisons?
> >
> > Besides, the missing regions in image inpainting are always with regular patterns. For example, in this paper, those regions are all black and square blocks. However, if such patterns become more diverse, is this method still effective?
> >
> > This may show the applicability of this method to more diverse applications like the specular high light observed on captured images. Thanks.

---

> > ### Comment · Reviewer_XsAp · 2022-12-07
> > **Thanks.**
> >
> > Thanks for the response. I have raised my scores a little bit. This is mainly due to the contribution in pixel-wise anoamly detection. For the image inpainting issue, I still think that the current framework still has a long way to go.

---

### Official Review · Reviewer_pBCc · 2022-10-24

**Confidence:** 3
**Correctness:** 3
**Technical Novelty And Significance:** 3
**Empirical Novelty And Significance:** 3
**Recommendation:** 6

**Clarity, Quality, Novelty And Reproducibility:**

The paper's idea is novel, but the experiement is not thorough, they presented limited cases in the paper.

**Strength And Weaknesses:**

Strength: 1. Paper is well written and easy to understand.
2. The research problem is important and has practical value.
3. Their proposed method seems novel.

Weakness: The experiment seems not thorough, For example, In Table 1, I'm curious about why they only compare limited cases in the experiment. I think the method should compare with more previous works and datasets to demonstrate its efficiency. Besides, I think the paper should add an ablation study to demonstrate the method.

**Summary Of The Paper:**

The paper presents a Robust GAN-inversion method for restoring gross corrupted images. They also extend RGI to Relaxed RGI for generator fine-tuning to mitigate the gap between the GAN learned manifold and the true image manifold. Their result achieves sota performance in two applications: mask-free semantic inpating and unsupervised pixelwise anomaly detection.

**Summary Of The Review:**

I think the paper's idea is novel, and the topic of this paper is also very important, I think the paper should add more experiments to demonstrate their method.

---

> ### Author Response · Authors · 2022-11-17
> **Response to Reviewer pBCc, Part I.**
>
> We deeply appreciate your valuable comments and constructive feedback. We have provided the code for reproducing the results, more implementation details can be found in the code. Modifications to the original manuscript have been highlighted in blue in the revised manuscript. Below we provide our point-by-point responses to your detailed remarks, which we hope can address your current reservations.
>
> ## Response to strength:
>
> We thank the reviewer for this comment. The proposed RGI/R-RGI method is a non-trivial extension of the traditional robust machine learning methods, such as robust principal component analysis (RPCA) [9], to the deep learning era, by introducing a learned deep generative prior (DGP). In this way, the proposed method can achieve good results in problems that are originally difficult by either using GAN-inversion or RPCA alone, such as mask-free semantic inpainting of complex human face images or unsupervised pixel-wise anomaly detection in complex background images from a real manufacturing line. As the result, our work is indeed related to many previous works, but surprisingly few are a fair comparison. We carefully designed the experiment study from from 3 different perspectives, including (i) following conventions of problem settings in the field, (ii) covering commonly used datasets, and (iii) comparing with baselines that can achieve a fair comparison. In addition, we even compare with several methods that do require more information.
>
> Below, we provide the details in point-by-point responses.
>
> ## Response to weakness:
> 1) **In terms of the two cases that we compared (Table 1), we follow the convention in GAN-inversion for semantic inpainting [1, 2].** The main reason is that we aim to demonstrate the robustness of the proposed method under sparse gross corruption. As can be seen in the problem formulation (Equation 3 and 4) there is no other assumptions related to mask shape. Therefore, block, random or other missing mechanisms can fulfill this goal as long as they are sparse. To address the reviewer’s concern, we added one more case of irregular missing masks (Irregular Mask Dataset [3]) in the revised paper. We repeat the same experiment and adopt a different irregular missing mask for each test image. Due to the space limit, we include the results in Appendix I. We can observe a consistent performance of changing from block missing to irregular shape missing, which validates our claim.
> 2) **In terms of previous works, we have compared with available literature satisfying the mask-free semantic inpainting goal.** The mask-free (in both the training and testing stage) image inpainting is a new problem. Therefore, there is limited literature that covers this goal. To provide a more comprehensive result, we even compare with some directly related literature [1, 4] that do require a preconfigured segmentation mask for semantic inpainting. We demonstrated that our proposed method, even though without the preconfigured segmentation mask requirement, can achieve comparable performance as [1, 4]. A detailed discussion on related literature can be found in the comprehensive literature review for the mask-free semantic inpainting task in Appendix A.1.
> A detailed discussion on related literature can be found in the comprehensive literature review for the mask-free semantic inpainting task in Appendix A.1.
> 3) **In terms of datasets, we have covered commonly used datasets for developing and benchmarking semantic inpainting algorithms [2, 4, 5] including CelebA[6], Stanford Cars[7] and LSUN [8].**

---

> > ### Author Response · Authors · 2022-11-17
> > **Response to Reviewer pBCc, Part II.**
> >
> > 4) In terms of the ablation study, we gently remark that there are indeed ablation studies in our paper. Due to the drawback of current GAN-inversion methods including (i) lack of robustness and (ii) gap between the approximated and actual manifolds, current GAN-inversion methods’ performance is limited on mask-free semantic inpainting and unsupervised pixel-wise anomaly detection applications. Therefore, we propose an RGI method to address the robustness issue and further generalize it to R-RGI to mitigate the GAN approximation gap: GAN-inversion -> RGI -> R-RGI. Therefore, the comparison between GAN-inversion and RGI is the ablation study for RGI; and the comparison between RGI and R-RGI is the ablation study for R-RGI.
> >     - In the semantic inpainting experiment: the Yeh et al. (2017) w/ mask baseline is directly applying the GAN-inversion to mask-free Semantic inpainting applications. Compared to Yeh et al. (2017) w/ mask baseline, the only modification in the RGI method is to learn a missing region mask during the inversion process. Other than that, they share the same optimization procedure as well as the dataset. Therefore, the comparison between Yeh et al. (2017) w/ mask baseline and RGI is an ablation study for RGI.
> > Compared to RGI, the only modification in R-RGI is to mitigate the GAN approximation gap by jointly optimizing the generator. Other than that, they share the same optimization procedure as well as the dataset. Therefore, the comparison between RGI and R-RGI is an ablation study for R-RGI.
> >     - In the unsupervised pixel-wise anomaly detection experiment: the AnoGAN baseline is directly applying the GAN-inversion to unsupervised pixel-wise anomaly detection applications. Compared to the AnoGAN baseline, the only modification in the RGI method is to learn a missing region mask during the inversion process. Other than that, they share the same optimization procedure as well as the dataset. Therefore, the comparison between the AnoGAN baseline and RGI is an ablation study for RGI.
> >
> > **References**
> >
> > [1]    	X. Pan, X. Zhan, B. Dai, D. Lin, C. C. Loy, and P. Luo, "Exploiting deep generative prior for versatile image restoration and manipulation," IEEE Transactions on Pattern Analysis and Machine Intelligence, 2021.
> >
> > [2]    	J. Gu, Y. Shen, and B. Zhou, "Image processing using multi-code gan prior," in Proceedings of the IEEE/CVF conference on computer vision and pattern recognition, 2020, pp. 3012-3021.
> >
> > [3]    	G. Liu, F. A. Reda, K. J. Shih, T.-C. Wang, A. Tao, and B. Catanzaro, "Image inpainting for irregular holes using partial convolutions," in Proceedings of the European conference on computer vision (ECCV), 2018, pp. 85-100.
> >
> > [4]    	R. A. Yeh, C. Chen, T. Yian Lim, A. G. Schwing, M. Hasegawa-Johnson, and M. N. Do, "Semantic image inpainting with deep generative models," in Proceedings of the IEEE conference on computer vision and pattern recognition, 2017, pp. 5485-5493.
> >
> > [5]    	W. Wang, L. Niu, J. Zhang, X. Yang, and L. Zhang, "Dual-path Image Inpainting with Auxiliary GAN Inversion," in Proceedings of the IEEE/CVF Conference on Computer Vision and Pattern Recognition, 2022, pp. 11421-11430.
> >
> > [6]    	Z. Liu, P. Luo, X. Wang, and X. Tang, "Deep learning face attributes in the wild," in Proceedings of the IEEE international conference on computer vision, 2015, pp. 3730-3738.
> >
> > [7]    	J. Krause, M. Stark, J. Deng, and L. Fei-Fei, "3d object representations for fine-grained categorization," in Proceedings of the IEEE international conference on computer vision workshops, 2013, pp. 554-561.
> >
> > [8]    	F. Yu, A. Seff, Y. Zhang, S. Song, T. Funkhouser, and J. Xiao, "Lsun: Construction of a large-scale image dataset using deep learning with humans in the loop," arXiv preprint arXiv:1506.03365, 2015.
> >
> > [9]	E. J. Candès, X. Li, Y. Ma, and J. Wright, "Robust principal component analysis?," Journal of the ACM (JACM), vol. 58, no. 3, pp. 1-37, 2011.

---

### Official Review · Reviewer_sD9p · 2022-10-24

**Confidence:** 4
**Correctness:** 4
**Technical Novelty And Significance:** 3
**Empirical Novelty And Significance:** 3
**Recommendation:** 6

**Clarity, Quality, Novelty And Reproducibility:**

The content of the article is relatively easy to understand. The method is not complicated and should be easily implemented.

**Strength And Weaknesses:**

Strength:
The problem is important. The solution is simple, and the results are good.
It's actually surprising that such a simple method can work. As far as I know, before the stylegan came along, this kind of method was very difficult to work.

Weaknesses:
1. The trade-off parameter $\lambda$ seems to be an important value, but the article does not show the sensitivity of the method to lambda.
2. The proof may not make much sense. This paper provides some theoretical guarantees. I briefly checked the proof. Its biggest problem is that some assumptions basically do not appear in real-world applications. But the final experimental result was good. Probably this does not experimentally affect the performance of the proposed method. But on the other hand, this reduces the contribution of this paper, because it seems that this guarantee is not important.

**Summary Of The Paper:**

This paper studies the GAN-Inversion problem. In particular, the GAN-inversion problem when the image suffers from missing or some pixels are no longer reliable. The proposed method is very simple and straightforward. A mask is introduced to the inversion optimization, and the mask has a constraint on the 0-norm. The results seem to be good, and this simple method works. The author also provided theoretical proof for this method.


**Summary Of The Review:**

My rating is based on: 1) there are no major technical omissions, 2) the problem studied is important, and 3) the proposed method works. But since 1) the method is simple, 2) the proof is of limited value. I don't rate it too high.

---

> ### Author Response · Authors · 2022-11-17
> **Response to Reviewer sD9p, Part I.**
>
> We deeply appreciate your valuable comments and constructive feedback. We have provided the code for reproducing the results, more implementation details can be found in the code. Modifications to the original manuscript have been highlighted in blue in the revised manuscript. Below we provide our point-by-point responses to your detailed remarks, which we hope can address your current reservations.
>
> ## Response to strength:
> This research is motivated by real applications but arrives at a simple problem formulation that works. The simplicity and intuitiveness are indeed the beauty of the proposed method. We are glad that the reviewer also noticed that. Yet such methodological simplicity and practical effectiveness are neither arbitrary nor unfounded -- it stems directly from the principles behind the development of the method. We would like to give a brief summary, which we hope to better clarify our contribution.
>
> Methodologically, the proposed RGI/R-RGI method is a non-trivial extension of the traditional robust machine learning methods, such as robust principal component analysis (RPCA) [2]. Traditional robust machine learning methods are popular robust signal restoration methods in the pre-deep-learning era, because of their simple and intuitive problem formulation but (somehow surprisingly) successful and wide application scenarios, such as including robust signal denoising and dimensional reduction [2], dynamic foreground and static background separation and anomaly detection [3, 4], as long as the signal satisfies the statistical assumptions of robust machine learning methods, such as low rank (RPCA), smoothness, and so on. However, we notice that in lots of applications, such assumptions can be restrictive. To address this issue, we introduce a learned deep generative prior (DGP), by GAN, to the traditional robust learning methods.
>
> By such a non-trivial extension, the proposed method not only inherits the robust nature, simple and intuitive problem formulation of robust machine learning, but also introduces a strong modeling capability. And in turn, we address the robustness and approximation gap in the traditional GAN-inversion method. Therefore, we name it Robust GAN-inversion (RGI).
>
> In this way, the proposed method can achieve good results in problems that are originally difficult by either using GAN-inversion or RPCA alone, such as mask-free semantic inpainting of complex human face images or unsupervised pixel-wise anomaly detection in complex background images from a real manufacturing line.
>
> ## Response to weakness #1:
> We thank the reviewer for pointing out the sensitivity analysis, and we added a sensitivity analysis in appendix J. The trade-off parameter $\lambda$.  is indeed important, which controls the sparsity level of the learned mask. Our result is not very sensitive to the trade-off parameter $\lambda$. This makes it is easy for parameter tuning, i.e., there is a plateau region around the optimal $\lambda^*$.  and thus selecting any $\lambda$ value around the $\lambda^*$ will give acceptable performance.

---

> > ### Author Response · Authors · 2022-11-17
> > **Response to Reviewer sD9p, Part II.**
> >
> > ## Response to weakness #2:
> > We appreciate that the reviewer read the theorems and checked the proof. We would like to address the reviewer’s concerns in the following three aspects: those assumptions are reasonable; our empirical results verify the validity of the Theorems; and the theorems (as well as the proofs) are important to the field of robust machine learning as well.
> > 1) **There are four assumptions in the proof, we would like to examine each of them to address the reviewer’s concern to show those assumptions are reasonable:**
> > - GAN learns an accurate image manifold, i.e., there exists $z^*$ such that $\|x-G(z^* )\|_0 \leq n_0$;
> > This assumption assumes that the data is generated from the model we proposed (or there is no GAN approximation gap). This assumption is easier to be satisfied when the GAN is trained on large-scale high-quality dataset. However, we agree that it can be strong in practice where the training data is limited for GAN, which will lead to a large GAN approximation gap. We took this into consideration and proposed an R-RGI method that can mitigate such an approximation gap when this assumption is not satisfied. This is another contribution of the paper.
> > - $z$ is bounded, or equivalently there exists $R>0$ such that $\|z\|_1 \leq R$, i.e., $z\in\mathbb{S}^d$ with $\mathbb{S}^d := [-R,R]^d$;
> > This assumption is just for the proof purpose, in practice, we can always find a large enough number $R$ that can bound the latent code $z$. Therefore, there is no gap in practice.
> > - $L_{rec}(\cdot)=\|\cdot\|_2^2$.
> > The $l_2$ loss is widely used in previous works such as \citep{yeh2017semantic}. In all our experiments, we also adopt an $l_2$ loss and demonstrated satisfactory results in both mask-free semantic inpainting and pixel-wise anomaly detection applications.
> > - $G(z)$ is continuous.
> > The continuous assumption of the generator network (continuous with respect to $z$) is also easy to satisfy. As a straightforward result from real analysis, the composition of continuous functions is still continuous. Since each layer of the network is continuous, the resulted $G(z)$ is continuous.
> > 2) **Moreover, our experiments verify the validity of the Theorems.**
> > - **The Theorems have been validated by a simulation study.** We have conducted a simulation study in Appendix C to validate the Theorem, where all assumptions are strictly satisfied, we can observe that the RGI demonstrates satisfactory performance as predicted by the Theorems.
> > - **The final experiment results reflect the Theorems.** In the final experiment, assumptions (ii)-(iv) are strictly satisfied. However, it is difficult to verify assumption (i) directly. In mask-free semantic inpainting, we can compare the performance difference (PSNR value is directly related to the result of the Theorems) between Yeh et al. (2017) w/ mask (or RGI) and Pan et al. (2021) w/ mask(or R-RGI), since the only difference between them are whether fine-tuning the generator to mitigate the GAN approximation gap. If the GAN approximation gap is small, fine-tuning the generator will not have significant performance improvement.
> >     - For example, as shown in Table 1, in CelebA experiments, the PSNR value difference between Yeh et al. (2017) w/ mask (or RGI) and Pan et al. (2021) w/ mask( or R-RGI) is small, which indicates a small GAN approximation gap (or assumption (i) is almost satisfied). In this case, we can observe a satisfactory performance of the RGI method, which indicates the experiment result indeed reflects the Theorems.
> >     - For another example, as shown in Table 2, in Cars experiments, the PSNR value difference between Yeh et al. (2017) w/ mask (or RGI) and Pan et al. (2021) w/ mask( or R-RGI) is large, which indicates a large GAN approximation gap (or assumption (i) is almost significantly violated), which is mainly because of the relatively small training set failed to cover all the variations in the data. The RGI method (as well as Yeh et al. (2017) w/ mask) has poor performance, which is expected. In this case, the proposed R-RGI method that mitigates the GAN approximation gap will significantly improve the result, which is one of our contributions.
> > 3) **The theorems (as well as the proofs) are important to the field of robust machine learning as well.** As we mentioned before, this work can be seen as an extension of the robust machine learning method in the deep learning era. The theorems (as well as the proofs) provide such a robustness guarantee, which is important content for robust machine learning as well.

---

> > > ### Author Response · Authors · 2022-11-17
> > > **Response to Reviewer sD9p, Part III.**
> > >
> > > **References**
> > >
> > > [1]    	X. Xia et al., "GAN-based anomaly detection: A review," Neurocomputing, 2022.
> > >
> > > [2]    	E. J. Candès, X. Li, Y. Ma, and J. Wright, "Robust principal component analysis?," Journal of the ACM (JACM), vol. 58, no. 3, pp. 1-37, 2011.
> > >
> > > [3]    	T. Bouwmans, A. Sobral, S. Javed, S. K. Jung, and E.-H. Zahzah, "Decomposition into low-rank plus additive matrices for background/foreground separation: A review for a comparative evaluation with a large-scale dataset," Computer Science Review, vol. 23, pp. 1-71, 2017.
> > >
> > > [4]    	T. Bouwmans and E. H. Zahzah, "Robust PCA via principal component pursuit: A review for a comparative evaluation in video surveillance," Computer Vision and Image Understanding, vol. 122, pp. 22-34, 2014.

---

### Official Review · Reviewer_fpcW · 2022-10-24

**Confidence:** 3
**Clarity, Quality, Novelty And Reproducibility:** The writing quality is good.
**Correctness:** 4
**Technical Novelty And Significance:** 3
**Empirical Novelty And Significance:** 2
**Recommendation:** 6

**Strength And Weaknesses:**

Strength:
- The paper is well-written and easy to follow. The two tasks are closely related and the proposed methods make sense.
- The two-stage optimization further closes the GAN gaps. The generalization of the method is intuitive.

Weaknesses:
- The idea is simple, effective and intuitive, but the problem setting is not that practical.
    (1) Center block is too easy to be overfitted, and it cannot reveal the advantages of the mask learning.
    (2) Random missing is also not practical enough, and a simple denoising method may be more effective in resolving the problem.
    (3) Synthetic defects are limited, and cannot be proven to be generalized to real tasks.
- The sparsity assumption is limited. Similar ideas can also be found in paper [1], while the authors of that inpainting paper proposed to jointly optimize the mask and reconstruction loss during training.
- It would be interesting to study whether we can learn the regional GAN-inversion by matching partial generated images with the ground truth. For example, whether we can optimize the inversion network to only extract the foreground objects or salient objects. It may be an interesting extension of this work.
- Finetunning the generator may be not that practical for real applications.

[1] Zeng, Yu, et al. "High-resolution image inpainting with iterative confidence feedback and guided upsampling." European conference on computer vision. Springer, Cham, 2020.

**Summary Of The Paper:**

The authors proposed a robust GAN-inversion method to cope with restoring images with unknown corruption and detecting unknown defects. They addressed the problem by involving the mask in the optimization process, and encouraging it to be sparse. Besides, to further close the GAN gap, they proposed to finetune the GAN generators while adapting new datasets. The authors showed the experimental results on both mask-free inpainting and unsupervised defect detection tasks, and claim it outperforms SOTA, and can be applied to other SOTA framework easily.

**Summary Of The Review:**

The major concerns of this paper:
- The synthetic dataset (for both tasks) fail to convince the reviewer that the proposed method is effective enough to recover the unknown corruption in the wild. If the author claimed only synthetic dataset can be used due to the lack of clean images for training GANs, then there are additional concerns on how the proposed methods can be generalized to more real-world tasks. Or the problems are not about the synthetic dataset, but the synthetic data (center block / random pixel missing) in this paper is not that realistic and diverse to reveal the advantages of the proposed method.
- The evaluation is poor, especially for the mask-free inpainting tasks. Baselines are weak, and the metrics are not advanced to demonstrate the visual quality. For GAN-based generation, at least LPIPS/FID should also be reported.
- More interesting topics like partial GAN-inversion should be explored to prove the robustness.

---

> ### Author Response · Authors · 2022-11-17
> **Response to Reviewer fpcW, Part I.**
>
> We deeply appreciate your valuable comments and constructive feedback. We have provided the code for reproducing the results, more implementation details can be found in the code. Modifications to the original manuscript have been highlighted in blue in the revised manuscript. Below we provide our point-by-point responses to your detailed remarks, which we hope can address your current reservations.
>
> ## Response to weakness #1:
> We thank the reviewer for this comment.
>
> **In mask-free semantic inpainting experiment, the use of block and random missing follows the convention in GAN-inversion for semantic inpainting [1, 2].** The main reason is that we aim to demonstrate the robustness of the proposed method under sparse gross corruption. As can be seen in the problem formulation (Equation 3 and 4) there are no other assumptions related to mask shape. Therefore, block, random or other missing mechanisms can fulfill this goal as long as they are sparse. Moreover, one advantage of GAN-based semantic inpainting methods [1-3] is that there is no overfitting issue towards any specific mask shape, since the training process is regular GAN training on a clean image data set that does not involve any mask information.
>
> To address the reviewer’s concern, we added one more case of irregular missing masks (Irregular Mask Dataset [4]) in the revised paper. We repeat the same experiment and adopt a different irregular missing mask for each test image. Due to the space limit, we include the results in Appendix I. We can observe a consistent performance of changing from block missing to irregular shape missing, which validates our claim.
>
> **In unsupervised pixel-wise anomaly detection experiment, we carefully create a synthetic dataset to best reflect the challenges and characteristics of anomaly detection in current mass production.**
>
> 1) **In terms of application, the synthetic defect dataset reflects the unique and new problem we want to solve:** (a) a large training set of clean images is cheap and easy to collect; (b) we require pixel-wisely  accurate annotations for evaluating the algorithm performance. This research is motivated by real applications in product cosmetic inspection of a major consumer electronics manufacturing company. For cosmetic inspection in nowadays mass production of high-stake products, there are two unique characteristics, (a) we require a pixel-wise anomaly characterization that is more challenging than current methods that mainly focus on sample level or localization level [5]; (b) collecting a large number of defect-free product images (on the magnitude of several thousand per day and even more) is cheap which provide opportunities for algorithm development (a data-rich but defect/label-rare environment). The proposed RGI/R-RGI method aims for this application. However, current publicly available anomaly detection datasets (such MVTec and BTAD) cannot reflect such challenges and opportunities, due to the small training set and annotation issue. Unfortunately, due to company policy, we are not able to share the internal dataset. Therefore, we provide the results on a synthetic dataset. This synthetic dataset, leveraging the advantage of various defective region masks from MVTec dataset and the relatively large number of clean training images in BTAD dataset, can provide a controllable environment, with pixel-wisely accurate ground truth defective region segmentation masks and a training set aligned with the problem setting. (the synthetic defect generation process can be found in Appendix F).
>
> 2) **In term of methodology, we mainly focus on solving the robustness issue and GAN approximation gap issue of GAN-inversion-based methods [3-6], which are designed for applications with a large dataset of cleaning training images.** In semantic inpainting applications, large scale public image datasets are available. We are able to demonstrate the proposed method on those datasets and comparing with available baselines. However, for unsupervised pixel-wise anomaly detection, there is no publicly available datasets supporting the development or benchmarking of GAN bassed anomaly detection methods, despite its popularity in the anomaly detection field [11]. In this case, we adopt a synthetic dataset to demonstrate our method. We would like to place the current work as an initial attempt to solve the unsupervised pixel-wise anomaly detection problem in a data-rich but defect/label-rare environment, which is seldom discussed in current anomaly detection literature. And hope we can raise the attention of researchers in this field on the important but less addressed problem.

---

> > ### Author Response · Authors · 2022-11-17
> > **Response to Reviwer fpcW, Part II.**
> >
> > ## Response to weakness #2:
> > We thank the reviewer for this comment. Here, we would like to address the reviewer’s concern in two aspects:
> > 1) **The main difference between our work and [6] (the citation #[1] in the reviewer's comment).** [6] iteratively updates the inpainted content inside the hole whose shape and location are known beforehand, to address the artifacts when dealing with large holes. However, we focus on restoring the image under unknown sparse gross corruptions (unknown relatively small holes in semantic inpainting). We agree that incorporating more advanced techniques, such as the Iterative Confidence Feedback and Guided Upsampling in [1], can be an interesting direction of future work in dealing with large missing regions. However, it is not the focus of this paper.
> > 2) **The sparsity assumption is principled and practical.**
> > - **Methodologically, sparsity of corruption is important to promote robustness.** As mentioned in the Discussion (Section 3.3.), the proposed RGI/R-RGI aims for robust signal restoration under sparse gross corruptions, which is a non-trivial extension of robust learning methods to the deep-learning era. Traditional robust machine learning methods, such as Robust Principal Component Analysis (RPCA), use statistical priors (low rank in RPCA) to model the signal to be restored and sparsity to model the corruptions, where sparsity assumption of the corruption is commonly used and important to promote robustness. The proposed RGI/R-RGI method aims to address the restrictive statistical priors in modeling the signal to be restored by introducing a learned deep generative prior (DGP) from GAN, but inherent RPCA’s sparsity assumption of corruptions to promote robustness. We demonstrate that the proposed method can solve problems that are originally difficult by either using GAN-inversion or RPCA alone, such as mask-free semantic inpainting of complex human face images or unsupervised pixel-wise anomaly detection in complex background images from a real manufacturing line.
> > - **Practically, the sparsity assumption of corruption is reasonable.** For semantic inpainting on CelebA dataset, we can inpainting up to 25% block missing regions and 50% random missing. However, inpainting extremely large missing holes is beyond the scope of this method. For anomaly detection, sparsity is a common situation in in-situ sensing for defect detection, where the defect area within an image is innately sparse for high-stake production lines.
> >
> > ## Response to weakness #3:
> > We thank the reviewer for pointing out this important and interesting direction. There are two different types of matching, including matching the background (corruption-free) region and the foreground (salient object) region, relating to different research topics. We mainly focus on matching the background region. Matching with (extracting) the foreground (salient object) region is out of our scope.
> >
> > 1) **In this research, we focus on solving the robustness issue of GAN-inversion with respect to unknown gross corruption, in which matching with the background (corruption-free) region is a natural setting that guides the learning of GAN-inversion.** It is indeed a common scenario in semantic image inpainting [1-3] (where the background corresponds to the unmissed region) and unsupervised anomaly detection (where the background corresponds to the defect-free region). In this paper, we move one step forward by learning the regional GAN-inversion and which region to match simultaneously, when a segmentation mask indicating the background region (or corrupted region) is not available beforehand, such as for mask-free semantic inpainting and unsupervised anomaly detection applications.
> > 2) **We can solve some foreground-background separation problems by subtracting the restored background from the input image (such as defect detection, where the defective region can be seen as foreground and the defective-free region can be seen as background).** This is the main idea for applying RPCA types of algorithms for foreground and background separation applications in the pre-deep-learning era [7]. However, we would like to acknowledge that foreground-background separation [8] (and salient object detection [9]) is a fairly large research field where extracting (matching) foreground (salient object) is indeed necessary and important for some applications. But considering the main problem that we aim to solve (robustness in GAN-inversion under unknown sparse gross corruptions), matching with (extracting) the foreground (salient object) region is out of our scope. Therefore, we leave it for an interesting future direction to explore.

---

> > > ### Author Response · Authors · 2022-11-17
> > > **Response to Reviwer fpcW, Part III.**
> > >
> > > ## Response to weakness #4:
> > > We thank the reviewer’s comment on generator fine-tuning in practice. We would gently remark that the method with fine-tuning (R-RGI) and without fine-tuning (RGI and GAN-inversion) are similar in terms of computational effort and data requirement. Therefore, the applicability of methods with and without fine-tuning are similar and both applicable in the application we studied.
> > > 1) **Fine-tuning the parameter is feasible (or can be done) in practice.** In both applications, the fine-tuning step DOES NOT require much additional effort compared to the RGI methods (the proposed method without fine-tuning):
> > > - In terms of optimization problem formulation, including generator fine-tuning is just adding another variable  (please see the discussion in Section 3.2)
> > > - In terms of computation, the same optimizer with the same number of gradient iterations is utilized. The only difference is that we optimize the variable $\theta$ in the last fixed number of iterations. Even though we need to evaluate the additional gradient of the additional variable $\theta$, we did not observe a significant difference in computational time between the method with fine-tuning (R-RGI) and without fine-tuning (RGI and GAN-inversion).
> > > - In terms of data requirement, the RGI and R-RGI require exactly the same data for training and testing.
> > > 2) **Generator finetuning has been successfully applied in real applications of semantic image inpainting [1]. We also demonstrate its applicability in unsupervised pixel-wise anomaly detection.** RGI/R-RGI for unsupervised pixel-wise anomaly detection is motivated by several real applications in product cosmetic inspection of a major consumer electronics manufacturing company. Generator fine-tuning is indeed the key step for GAN-inversion-based anomaly detection algorithms to work in practice.
> > >
> > > ## Response to the “summary of the review,  major concerns #2" (notice that other points have been covered in responses to weaknesses)
> > > We thank the reviewer for the comments. We would like to address the reviewer’s concerns in two aspects:
> > > 1) **In terms of baselines, we have compared with available literature satisfying the mask-free semantic inpainting goal.** The mask-free (in both the training and testing stage) image inpainting is a new problem. Therefore, there is limited literature that covers this goal. To provide a more comprehensive result, we even compare with some directly related literature [1, 3] that do require a preconfigured segmentation mask for semantic inpainting. We demonstrated that our proposed method, even though without the preconfigured segmentation mask requirement, can achieve comparable performance as [1, 3]. A detailed discussion on related literature can be found in the comprehensive literature review for the mask-free semantic inpainting task in Appendix A.1.
> > >
> > > 2) **In terms of metrics, we follow the convention in semantic image inpainting of comparing the SSIM/PSNR (including GAN-based methods [1, 3] and the reference mentioned by the reviewer [6]).** For semantic inpainting applications (and more broadly, signal restoration), we mainly aim to restore the corrupted signal, instead of generative a signal that looks real (in terms of a good LPIPS/FID score). Since the ground truth image that we aim to restore is available, we can directly evaluate the algorithm performance by calculating the similarity between the ground truth image and the restored one.

---

> > > > ### Author Response · Authors · 2022-11-17
> > > > **Response to Reviwer fpcW, Part IV.**
> > > >
> > > > **References**
> > > >
> > > > [1]    	X. Pan, X. Zhan, B. Dai, D. Lin, C. C. Loy, and P. Luo, "Exploiting deep generative prior for versatile image restoration and manipulation," IEEE Transactions on Pattern Analysis and Machine Intelligence, 2021.
> > > >
> > > > [2]    	J. Gu, Y. Shen, and B. Zhou, "Image processing using multi-code gan prior," in Proceedings of the IEEE/CVF conference on computer vision and pattern recognition, 2020, pp. 3012-3021.
> > > >
> > > > [3]    	R. A. Yeh, C. Chen, T. Yian Lim, A. G. Schwing, M. Hasegawa-Johnson, and M. N. Do, "Semantic image inpainting with deep generative models," in Proceedings of the IEEE conference on computer vision and pattern recognition, 2017, pp. 5485-5493.
> > > >
> > > > [4]    	G. Liu, F. A. Reda, K. J. Shih, T.-C. Wang, A. Tao, and B. Catanzaro, "Image inpainting for irregular holes using partial convolutions," in Proceedings of the European conference on computer vision (ECCV), 2018, pp. 85-100.
> > > >
> > > > [5]    	K. Roth, L. Pemula, J. Zepeda, B. Schölkopf, T. Brox, and P. Gehler, "Towards total recall in industrial anomaly detection," in Proceedings of the IEEE/CVF Conference on Computer Vision and Pattern Recognition, 2022, pp. 14318-14328.
> > > >
> > > > [6]    	Y. Zeng, Z. Lin, J. Yang, J. Zhang, E. Shechtman, and H. Lu, "High-resolution image inpainting with iterative confidence feedback and guided upsampling," in European conference on computer vision, 2020: Springer, pp. 1-17.
> > > >
> > > > [7]    	T. Bouwmans and E. H. Zahzah, "Robust PCA via principal component pursuit: A review for a comparative evaluation in video surveillance," Computer Vision and Image Understanding, vol. 122, pp. 22-34, 2014.
> > > >
> > > > [8]    	T. Bouwmans, A. Sobral, S. Javed, S. K. Jung, and E.-H. Zahzah, "Decomposition into low-rank plus additive matrices for background/foreground separation: A review for a comparative evaluation with a large-scale dataset," Computer Science Review, vol. 23, pp. 1-71, 2017.
> > > >
> > > > [9]    	A. Borji, M.-M. Cheng, Q. Hou, H. Jiang, and J. Li, "Salient object detection: A survey," Computational visual media, vol. 5, no. 2, pp. 117-150, 2019.
> > > >
> > > > [10]	E. J. Candès, X. Li, Y. Ma, and J. Wright, "Robust principal component analysis?," Journal of the ACM (JACM), vol. 58, no. 3, pp. 1-37, 2011.
> > > >
> > > > [11]	X. Xia et al., "GAN-based anomaly detection: A review," Neurocomputing, 2022.

---

> ### Comment · Reviewer_fpcW · 2022-11-23
> **Thanks for the detailed rebuttal**
>
> Thanks for the very detailed and comprehensive discussions regarding my concerns, and they are very helpful.
> After reading it, I feel the exploration in this paper is interesting, and the proposed methods also make a lot more sense.
>
> My main concern about generalization ability may not be easily addressed and verified. It seems there is only one visual example in Figure 9 showing the model performance on real mask-free detection and inpainting tasks. I still think the mask contents in the synthetic dataset can be too easy for the optimization problem. For example, as in Figure 10, the pixel values filled in the mask are constant, which are the average of the pixel content values. In real images, it won't be that ideal. To verify the robustness, it is still worth testing on more challenging real cases and showing the visualization. For training and testing on synthetic data, a two stage model like segmentation learning + inpainting could be possibly faster and achieves better quality and generalization ability (not sure but better to test). The goal of this paper should be showing the advantages of the optimization-based approach over those trivial two-stage pipelines in some additional ablation studies.
>
> However, the performance of the proposed method is limited by the domain of the GAN model, so it's not easy to directly make it work on images with domain gap. Finetunning makes it better, but how large the domain gap can be tolerated for finetuning? It seems there are many unknown problems while applying the model to practical applications.
>
> In summary, I no longer have concerns on the ideas in this paper, and the formulation of the problems. But the concerns on the practicability are not well addressed since the authors did not provide more visualization or domain gap analysis. When the abnormal pixels are too different from the background pixels, matching background using GAN-inversion and optimization may not be a too challenging question. Therefore, I also have some doubts about the direction proposed in the paper to address the mask-free inpainting problem.
>
> I may want to see some opinions from other reviewers.

---

### Official Review · Reviewer_TJ5g · 2022-10-26

**Confidence:** 3
**Correctness:** 3
**Technical Novelty And Significance:** 3
**Empirical Novelty And Significance:** 2
**Recommendation:** 6

**Clarity, Quality, Novelty And Reproducibility:**

The formulation of the proposed RGI and R-RGI methods is clear, but the implementation details are not clear enough and the limitations of the proposed methods are not discussed in the paper.

I have concerns with the quality of the paper in terms of the presentation and experimental justification.

The novelty of the paper seems solid mainly because this paper presents new extended formulation for GAN inversion with nice theoretical proofs.

This paper is not easy to reproduce since it doesn't provide implementation details on the GAN model training amd the network architecture of the GAN model used in the experiments.

**Strength And Weaknesses:**

Strengths:
1. This paper resents a robust GAN Inversion (RGI) method and a relaxed RGI (R-RGI) method for mask-free image inpainting and anomaly localization problems.
2. This paper presents two theorems for robust RGI that provides the theoretical foundation for the optimization formulation of the proposed algorithm.
3. Experimental results by applying the proposed RGI and R-RGI methods to two application problems on some public datasets are shown with promising results.

Weaknesses:
1. The proposed RGI or R-RGI methods require to solve the large optimization problems during the inference. There is no discussion on the computational cost involved in solving the large optimization problems during the inference. It could be very computationally infeasible for practical application.

2. Another limitation with the GAN inversion approach is that it requires a large number of normal images to train a GAN model for the specific task. In many cases, there are not enough numbers of normal images for training the GAN models. In their experiments on image inpainting, the datasets used in the experiments are CelebA, Standard cars, and LSUN bedroom all contain very large numbers of images for the GAN model training. But for MVTec and BTAD datasets, the numbers of training images are not a lot. There is no discussion on the numbers of images used in the model training for the implementation on anomaly detection experiments. There should be more discussion on the issue of training data requirement for the proposed method.

3. The experiments on pixel-wise anomaly detection are only performed on their synthetic defect dataset based on BTAD dataset. The experimental evaluation is not sufficient to demonstrate the proposed RGI or R-RGI achieve SOTA performance for anomaly detection. They should perform experiments with the standard protocol on the commonly used datasets, such as MVTec and BTAD datasets and include experimental comparison with some recent SOTA methods.

4. The discussion in section 3.3 claims the proposed RGI method is connected to robust statistics and the loss function in eq. 3 can be simplified by using the robust error function, which avoids introducing M in the loss function. However, there is no justification of this claim. If this is the case, is the resulting optimization problem also simpler to solve?

**Summary Of The Paper:**

This paper presents a robust GAN-Inversion (RGI) method for mask-free image inpainting and unsupervised pixel-wise anomaly detection. Furthermore, they further propose a relaxed RGI method for achieving better results. Promising experimental results on image inpainting and anomaly detection by using the proposed method are shown with comparison with some previous methods.

**Summary Of The Review:**

In summary, the paper has nice novelty in tern of the proposed new algorithms for solving the mask-free semantic inpainting and pixel-level anomaly detection problems. The proposed methods are accompanied by two theorems for robust GAN-Inversion. However, there are some concerns with the paper in terms of lack of discussion of the critical  limitations for the GAN-Inversion approach and the insufficient experimental justification, especially on the anomaly detection experiment.

---

> ### Author Response · Authors · 2022-11-17
> **Response to Reviewer TJ5g, Part I.**
>
> We deeply appreciate your valuable comments and constructive feedback. We have provided the code for reproducing the results, more implementation details can be found in the code. Modifications to the original manuscript have been highlighted in blue in the revised manuscript. Below we provide our point-by-point responses to your detailed remarks, which we hope can address your current reservations.
>
> ## Response to weakness #1:
> We agree with the reviewer’s insightful observation that computational cost can be a problem for practical applications when online inference is needed. Therefore, we integrated a discussion of the computation cost in Appendix H.1.
>
> However, we want to emphasize that this paper is the first attempt of discussing and improving the robustness of GAN-inversion borrowing the merit from the traditional robust machine learning field [1, 2], and in turn generalize the traditional robust machine learning to a wider range of applications.
>
> As an initial attempt, we focus on solving the robustness issue and GAN approximation gap issue for the GAN-inversion. Therefore, we study the robust GAN-inversion for the optimization-based method, which is most popular due to its superior image restoration quality compared to learning-based inversion methods, even though it is indeed computationally expensive (also noticed by researchers in this field [3], a detailed discussion can be found in Section 2). Generalizing to learning-based as well as hybrid GAN-inversion methods for better computational efficiency is an immediate extension and we leave it for future work.
>
> ## Response to weakness #2:
> We appreciate the reviewer’s observation on the training image requirement of the proposed method.
>
> The number of images required by our method may depend on the task. In general, several thousand images are enough for R-RGI in manufacturing anomaly detection applications. In our case study, we use 900 images in training the GAN and demonstrated satisfactory performance.
>
> We also appreciate that the reviewer also noticed the training data requirement issue for the current GAN-based anomaly detection method. We also gave a discussion in Appendix H.2 on the issue of training data requirement, hope that can address the reviewer’s concern:
> 1) The proposed R-RGI method can reduce the training data requirement for GAN-based anomaly detection methods [3], which is also one of the major contributions of our paper. Training data requirement is a common challenge for GAN-based anomaly detection. We also notice this problem when we first adapt the current GAN-inversion-based methods [4-6] for anomaly detection, where the training data requirement is indeed a big problem and leads to their poor performance (please refer to the Appendix A.2 for a detailed discussion). In general, a small training set will lead to a large GAN approximation gap between the GAN learned manifold and the real image manifold of the test data. Therefore, for current GAN-inversion-based methods, there is still a large mismatch between the closest image in the learned image manifold and the ground trth background of the input test image, beyond the robustness issue. The R-RGI method achieved a significant performance improvement over current GAN-inversion-based anomaly detection methods in all 4 different defect scenarios that outperform the SOTA method (DRAEM) on this task (see Table 2 in Section 4.2). This demonstrates the effectiveness and necessity of the R-RGI method to mitigate such a gap caused by data issues.
> 2) We managed to reduce the training data from hundreds of thousands to several thousand, which is acceptable in our application. This research is motivated by several real applications in product cosmetic inspection of a major consumer electronics manufacturing company that has a data-rich but defects/label-rare environment, which is common for nowadays mass production. In such an environment, it is cheap to collect a large number of defect-free product images (on the magnitude of several thousand per day and even more), while expensive and time-consuming to collect and annotate defective samples due to the super-high yield rate and expert annotation requirements. We mainly aim to develop an unsupervised pixel-wise anomaly detection algorithm that can succeed in such an environment. Therefore, an extremely small training set is not within our scope.

---

> > ### Author Response · Authors · 2022-11-17
> > **Response to Reviewer TJ5g, Part II.**
> >
> > ## Response to weakness #3:
> > We thank the reviewer for this comment. Standard usage of current publicly available anomaly detection datasets (such MVTec and BTAD) does not satisfy our need, but we carefully create a setup to best reflect the challenges and characteristics of anomaly detection in the production scenario, and we carefully draw conclusions. We would like to give several reasons for using a synthetic defect dataset, and we hope can address your current reservations:
> > 1) Current publicly available anomaly detection datasets (such MVTec and BTAD) cannot reflect the challenges and opportunities for unsupervised pixel-wise anomaly detection in nowadays mass production. For cosmetic inspection in nowadays mass production of high-stake products, there are two unique characteristics, (a) we require a pixel-wise anomaly characterization that is more challenging than current methods that mainly focus on sample level or localization level [7]; (b) collecting a large number of defect-free product images (on the magnitude of several thousand per day and even more) is cheap which provide opportunities for algorithm development. The proposed RGI/R-RGI method aims for this application.  However, current publicly available anomaly detection datasets (such MVTec and BTAD) cannot reflect such challenges and opportunities, due to the small training set and annotation issue. Unfortunately, due to company policy, we are not able to share the internal dataset. Therefore, we provide the results on a synthetic dataset. This synthetic dataset, leveraging the advantage of various defective region masks from MVTec dataset and the relatively large number of clean training images in BTAD dataset, can provide a controllable environment, with pixel-wisely accurate ground truth defective region segmentation masks and a training set aligned with the problem setting. (the synthetic defect generation process can be found in Appendix F).
> > 2) We mainly focus on solving the robustness issue and GAN approximation gap issue of GAN-inversion-based methods [3-6], which are designed for applications with a large dataset of cleaning training images. Even though the R-RGI method can reduce the training data requirement by mitigating the GAN approximation gap caused by limited training clean data. Working on extremely small datasets, such as MVTec (200~300 training images for each category) is still out of GAN-based methods’ scope (please refer to Appendix E for a detailed discussion).
> >
> > We appreciate the reviewer’s observation that the result on the current synthetic dataset may not cover every aspect of challenges for pixel-wise anomaly detection. One of our another ongoing effort is to open source a large-scale synthetic cosmetic inspection image dataset that better reflects the challenges and opportunities faced by the manufacturing process. We would like to place the current work as an initial attempt to solve the unsupervised pixel-wise anomaly detection problem in a data-rich but defect/label-rare environment, which is seldom discussed in current anomaly detection literature. And hope we can raise the attention of researchers in this field on the important but less addressed problem.
> >
> > ## Response to weakness #4:
> > We thank the reviewer for this comment. We would like to address the reviewer’s concern in two aspects:
> > 1) The derivation is in Appendix B.1 (the proof of Theorem 1) and the simplified loss function is Equation 6. We apologize for the confusion caused by the presentation. To avoid confusion to readers, we added an explanation in the Connection to robust statistics in Section 3.3 in the revised paper.
> > 2) The resulting optimization problem is simpler to solve when adopting a separable loss function, but in general, introducing $M$ is necessary. The proposed method can be simplified when we are adopting a separable reconstruction loss function, such as $l_2$ loss, which can be written as the sum of the reconstruction loss of each pixel. In this case, it reduces to robust statistics and the simplified loss does make the optimization problem easier to solve. However, in general image processing and GAN-inversion, there are loss functions demonstrating better performance that may not be pixel-wisely separable, such as perceptual loss, discriminator loss, and so on. In this case, introducing the mask $M$ is necessary for incorporating robustness. As mentioned in Section 3.3, we aim to clarify that the proposed method not only can be simplified into similar loss functions as robust statistics in special cases, thus inheriting robustness properties, but is also flexible enough to incorporate robustness to reconstruction loss functions beyond convex formulations, such as the perceptual loss and discriminator loss.

---

> > > ### Author Response · Authors · 2022-11-17
> > > **References**
> > >
> > > **References**
> > >
> > > [1]    	C. Caramanis, S. Mannor, and H. Xu, "14 robust optimization in machine learning," Optimization for machine learning, p. 369, 2012.
> > >
> > > [2]    	E. J. Candès, X. Li, Y. Ma, and J. Wright, "Robust principal component analysis?," Journal of the ACM (JACM), vol. 58, no. 3, pp. 1-37, 2011.
> > >
> > > [3]    	X. Xia et al., "GAN-based anomaly detection: A review," Neurocomputing, 2022.
> > >
> > > [4]    	T. Schlegl, P. Seeböck, S. M. Waldstein, G. Langs, and U. Schmidt-Erfurth, "f-AnoGAN: Fast unsupervised anomaly detection with generative adversarial networks," Medical image analysis, vol. 54, pp. 30-44, 2019.
> > >
> > > [5]    	T. Schlegl, P. Seeböck, S. M. Waldstein, U. Schmidt-Erfurth, and G. Langs, "Unsupervised anomaly detection with generative adversarial networks to guide marker discovery," in International conference on information processing in medical imaging, 2017: Springer, pp. 146-157.
> > >
> > > [6]    	H. Zenati, C. S. Foo, B. Lecouat, G. Manek, and V. R. Chandrasekhar, "Efficient gan-based anomaly detection," arXiv preprint arXiv:1802.06222, 2018.
> > >
> > > [7]	K. Roth, L. Pemula, J. Zepeda, B. Schölkopf, T. Brox, and P. Gehler, "Towards total recall in industrial anomaly detection," in Proceedings of the IEEE/CVF Conference on Computer Vision and Pattern Recognition, 2022, pp. 14318-14328.

---

### Decision · Program_Chairs · 2023-01-20

**Decision:**

Accept: poster

**Justification For Why Not Higher Score:**

It is a borderline paper.

**Justification For Why Not Lower Score:**

Some reviewers expressed concerns on the novelty of the work, and the limitations of the method mentioned above.

**Metareview: Summary, Strengths And Weaknesses:**

The paper presents a robust GAN-Inversion (RGI) method for mask-free image inpainting and unsupervised pixel-wise anomaly detection. The paper also presents a relaxed RGI method for better results. The paper includes experimental results for image inpainting and anomaly detection, showing promising results compared to previous methods. Reviewers think the method is overall simple and effective. However, the paper lacks discussion of the computational cost involved in solving the large optimization problems required for the proposed methods (in responses the authors admitted the computational cost can be an issue for online inference), and the experiments can also benefit from more analysis (such as for the domain-gap in anomaly detection). Other potential limitations of the paper include the need for a large number of normal images to train the GAN model, and the lack of discussion on the number of images used for training in the anomaly detection experiments. Overall, this is a borderline paper, but reviewers think it may bring some additional useful discussions to the community, so I'm leaning towards acceptance.

**Note From Pc:**

if the above contains the word "oral" or "spotlight" please see: "oral" presentation means -> notable-top-5% and "spotlight" means -> notable-top-25%. As stated in our emails, we are disassociating presentation type from AC recommendations

**Summary Of Ac-Reviewer Meeting:**

Due to the distribution of reviewers in different time zones, it is challenging for us to find a good time to meet. Plus there are quite sufficient amount of author-reviewers, and reviewers-AC discussion happening online.